# Depot-specific mRNA expression programs in human adipocytes suggest physiological specialization via distinct developmental programs

**Heather J. Clemons[1,2], Daniel J. Hogan** [1,2¤a]*, **Patrick O. Brown**[1,2¤b]*

**1** Department of Biochemistry, Stanford University School of Medicine, Palo Alto, California, United States of America, **2** Howard Hughes Medical Institute, Stanford University School of Medicine, Palo Alto, California, United States of America

¤a Current address: Alpine Bio South San Francisco, San Francisco, California, United States of America
¤b Current address: Department of Biochemistry (Emeritus), Stanford University School of Medicine, Stanford, California, United States of America
* hogandj@alumni.stanford.edu (DJH); pbrown@stanford.edu (POB)

**Data Availability Statement:** Microarray data from this study are available in GEO GSE25194. Datasets

## Abstract

Adipose tissue is distributed in diverse locations throughout the human body. Not much is known about the extent to which anatomically distinct adipose depots are functionally distinct, specialized organs, nor whether depot-specific characteristics result from intrinsic developmental programs, as opposed to reversible physiological responses to differences in tissue microenvironment. We used DNA microarrays to compare mRNA expression patterns of isolated human adipocytes and cultured adipose stem cells, before and after *ex vivo* adipocyte differentiation, from seven anatomically diverse adipose tissue depots. Adipocytes from different depots display distinct gene expression programs, which are most closely shared with anatomically related depots. mRNAs whose expression differs between anatomically diverse groups of depots (e.g., subcutaneous vs. internal) suggest important functional specializations. These depot-specific differences in gene expression were recapitulated when adipocyte progenitor cells from each site were differentiated *ex vivo*, suggesting that progenitor cells from specific anatomic sites are deterministically programmed to differentiate into depot-specific adipocytes. Many developmental transcription factors show striking depot-specific patterns of expression, suggesting that adipocytes in each anatomic depot are programmed during early development in concert with anatomically related tissues and organs. Our results support the hypothesis that adipocytes from different depots are functionally distinct and that their depot-specific specialization reflects distinct developmental programs.

## Introduction

Fat depots are found at diverse sites throughout the human body; some in obvious, familiar places, like hips, thighs and abdomen, and others at less familiar sites, such as behind the

used to generate main figures will be available as supplemental material.

**Funding:** This work was supported by the Howard Hughes Medical Institute and by a grant from the NIH to POB, "Extending and Interpreting Molecular Portraits of Cancer" (NIH 5 R01 CA111487). HJC was partially supported by the Stanford Genome Training Program (Grant Number 5 T32 HG000044).

**Competing interests:** The authors have declared that no competing interests exist.

eyeball (orbital fat pad), surrounding the heart (pericardial fat), under the kneecaps and protecting the spinal column [1, 2]. Fat covers the entire body, just beneath the skin (subcutaneous fat), and is also found deeper inside the body surrounding our guts (e.g., omental fat) and delicate organs such as the kidneys.

In mammals, two types of fat tissue with distinct developmental lineages, molecular profiles, and functions have been defined, brown adipose tissue and white adipose tissue [3, 4]. In adult humans brown adipose tissue is confined to depots in the upper chest and neck while the majority of depots comprise white adipose tissue [5]. White adipose tissue plays integral roles in energy metabolism and storage, mechanical support, paracrine and endocrine signaling, as well as immunological, reproductive and aesthetic functions. Adipose tissue is composed of diverse cell types including adipocytes, but also fibroblasts, smooth muscle cells, mesenchymal stem cells, endothelial cells, blood cells, immune cells and neurons [4]. Within some fat depots there are cells with characteristics of both brown and white adipocytes, called beige adipocytes [6].

All fat is not created equal; anatomically distinct fat depots contribute differently to adipose tissue functions, and respond differently to physiological and genetic variations. Perhaps most obvious are differences among fat depots in their capacity for lipid storage; no matter how much we overeat, we do not gain much weight in the palms of our hands, the fat depots behind our eyeballs, or our eyelids. Anatomic depot-specific differences in adipose tissue response to positive energy balance are deeply intertwined with susceptibility to obesity-associated diseases [7]. An "apple-shaped" pattern of fat deposition (predominantly above the hips) is associated with a significantly higher risk of type 2 diabetes and cardiovascular disease as compared to a "pear-shaped" (predominantly below the waist) fat distribution pattern. Accumulation of fat predominantly inside the abdominal cavity relative to abdominal subcutaneous depots has been strongly linked to metabolic syndrome, potentially due to differences in immune and endocrine functions, insulin sensitivity and lipid metabolism [7, 8].

Differences among fat depots are further highlighted by mutations and responses to drugs with site-specific effects. Congenital lipodystrophies and Graves-associated ophthalmopathy each lead to characteristic patterns of expansion or contraction of specific anatomic fat depots [9, 10]. Congenital lipodystrophies encompass a range of disorders and underlying mutations with diverse depot-specific effects. Likewise, treatment with HIV protease inhibitors, elevated glucocorticoid levels and even puberty in females have distinct site-specific effects on fat composition [11]. Thus, a given disease or physiological stimulus can elicit different responses from adipose tissue located at different sites in the body.

Among human fat depots, the omental (internal/intra-abdominal/peritoneal/visceral), abdominal (subcutaneous), and, to a lesser extent, thigh (subcutaneous) adipose depots have thus far been the focus of the majority of studies due to the differential associations of expansion of these depots with type 2 diabetes and metabolic syndrome. Notable differences have been reported in capacities for several metabolic, endocrine and immune functions (for example, [12–20]) providing molecular clues underlying their physiological specialization. While numerous functional differences among adipocytes from different depots have been reported, how and when these differences are acquired and maintained through specific gene expression programs is largely unknown. How do the gene expression programs in adipocytes vary as a function of anatomical location and among individuals and what might those differences tell us about the intrinsic and extrinsic cues that drive their divergent specialization?

There is evidence that functional differences among fat depots are, at least, in part developmentally programmed [3, 21]. Studies in mice comparing mRNA expression between adipocytes and preadipocytes isolated from the visceral epididymal and subcutaneous abdominal depots, and human studies comparing visceral omental to abdominal subcutaneous fat depots,

identified scores of mRNAs differentially expressed in fat cells from the respective depots [22, 23]. Several of these mRNAs encode developmental transcription factors whose expression differences persist after several passages in culture, suggesting they are epigenetically determined. Overexpression of one of these factors, homeobox protein TBX15, in a mouse preadipocyte model impairs adipogenesis [24]. In humans, TBX15 expression in omental fat negatively correlates with waist-to-hip ratio, a proxy for propensity for type 2 diabetes [22]. The reported anatomic pattern of TBX15 expression is strikingly different between mouse and human adipocytes; while TBX15 is specifically expressed in subcutaneous abdominal adipocytes in mice, it is preferentially expressed in omental adipocytes in humans.

Subsequent studies have expanded on adipose depot-specific expression of genes with roles in development, identifying differentially expressed mRNAs encoding proteins implicated in development, in adipocytes and/or preadipocytes isolated from two to five depots in mice or humans [25, 26]. A study comparing mRNA expression in human subcutaneous abdominal vs. subcutaneous gluteal adipose tissue found nine homeobox genes preferentially expressed in abdominal fat and two homeobox genes preferentially expressed in gluteal fat [26]. depot-specific expression was independent of Body Mass Index (BMI) and maintained following *ex vivo* differentiation. Genome-wide association studies (GWAS) identified several variants of developmental regulatory genes that correlate with obesity, including in TBX15 and HOXC13 [27–29]. A locus within an intron in the FTO gene, whose variants are reproducibly associated with obesity and type 2 diabetes, is actually an enhancer for the mesodermal developmental transcription factor IRX3 [30]. IRX3 null mice were ~25% lighter due to reduction in fat mass; however, depot-specific effects were not investigated [30, 31].

There is scant information regarding the embryologic origin and development of adipose tissue. During mouse and porcine fetal development, early adipose organs first appear as vascular structures with surrounding connective tissue [32–35]. The source(s) of mesenchymal progenitors recruited to these nascent organs has not been fully established, but recent lineage tracing experiments in mice affirm that mesodermal subtypes are the predominant source, with retroperitoneal and interscapular adipocytes deriving from paraxial mesoderm progenitor cells and subcutaneous triceps and inguinal and visceral mesenteric adipocytes originating from lateral plate mesoderm progenitor cells [36, 37]. Female perigonadal adipocytes derive from lateral plate mesoderm but male perigonadal adipocytes originate from paraxial mesoderm. Lineage tracing work in mice further suggests visceral adipocytes derive from splanchnic lateral plate, while subcutaneous adipocytes originate from somatic lateral plate [37–39]. Another study suggests that in mice salivary gland white adipocytes arise from the neural crest lineage [40]. Thus, it appears that adipocyte depots develop from multiple distinct lineages during embryonic development.

The abundance of developmental programming genes associated with patterns of obesity along with the recent lineage tracing experiments suggest that different adipose depots are developmentally distinct at multiple levels (e.g., head/neck vs. abdomen, visceral vs. subcutaneous). However, comparisons of gene expression programs in adipocytes have generally been limited to a few adipose depots (often in mice) and/or measurement of relatively few genes. The full extent to which the various human adipose depots are functionally and developmentally distinct organs therefore remains to be elucidated. How does expression of developmental transcription factors in adult adipose depots relate to their expression or known functions during early development?

To provide a more comprehensive and higher-resolution picture of the specialization of anatomically distinct adipose depots in humans, we profiled mRNA expression patterns in 83 adipose tissue specimens from 13 different depots; we also profiled expression patterns in separated adipocyte and stromal-vascular cell fractions, from 60 of these specimens, representing

seven depots. To assess whether site-specific features of adipocyte gene expression are dictated by a stable, intrinsic program or, alternatively, reflect a response to different *in vivo* microenvironments, we included ~50 adipose stem cell (ASC) samples isolated from five depots, cultured and differentiated *ex vivo*.

## Results

### Gene expression profiles reflect diversity among human adipose depots

We profiled 83 human adipose tissue samples from 13 different anatomic sites and 65 different donors. From among this set we chose to focus on internal or "visceral" adipose tissues at intra-abdominal/peritoneal sites (omental and pericolonic depots), in the retroperitoneal space, and subcutaneous adipose tissue from the female breast, abdomen, thigh and lower leg (S1 Fig). This set therefore includes the most-studied depots, subcutaneous abdominal and visceral omental fat, as well as five additional sites that are anatomically and morphologically diverse and readily obtainable from common surgeries. To protect patient privacy, we obtained only limited information on the donors, including age, sex, and in many cases, type of procedure (S1 Table). To minimize potentially confounding pathophysiological, genetic and gender influences, for each site, we analyzed samples from 5–12 different individuals, including both male and female donors.

To disentangle the expression programs of adipocytes from those of other cells in the tissue, we isolated adipocytes from 60 adipose tissue samples using an established method that takes advantage of the relatively low density of adipocytes (Fig 1A) (see Materials and methods).

Following this tissue fractionation ASCs were cultured from the pelleted stromal-vascular cell fraction and differentiated into adipocytes *ex vivo* (see Materials and methods). Total RNA was isolated from the "floated" adipocyte/lipid fraction, the pelleted stromal-vascular cells, cultured ASCs before and following differentiation, and untreated whole adipose tissue, respectively. Isolated RNA was amplified, fluorescently labeled, and analyzed by comparative hybridization to DNA microarrays (see Materials and methods, Fig 1A, S1–S4 Datasets).

### Adipose tissue fractionation enriches for adipocytes and stromal-vascular cells

We started with a broad examination of global mRNA expression in adipose tissue fractions. From the >15,000 well-measured mRNAs, we selected 2,809 distinct mRNAs whose relative abundance varied by at least one standard deviation from its mean across samples. Unsupervised hierarchical clustering of the mRNA expression profiles split the samples into three groups, each of which overwhelmingly comprised samples from the same cell fraction–adipose, stromal-vascular or total tissue, respectively (Fig 1B, S5 Dataset). Thus, the cell fraction from which the mRNA originated had a larger influence on variation in the overall pattern of expression than either the anatomic depot or donor. The mRNA expression patterns segregated into five main clusters. Cluster 3 included 590 mRNAs whose relative expression was higher in adipocytes compared to total tissue and stromal-vascular cells. This set of mRNAs included adipocyte markers (e.g., GPAM/GPAT1 Fig 1C [41]) and an abundance of mRNAs encoding proteins involved in "organic acid metabolic process" (117, $p < 1e^{-33}$, hypergeometric distribution function with Benjamini-Hochberg p-value adjustment), including "fatty acid metabolic process" (54, $p < 1e^{-19}$) and "carbohydrate metabolic progress" (47, $p < 1e^{-7}$) (Fig 1E). These results suggest that our tissue fractionation procedure reproducibly enriched for adipocytes. In contrast, cluster 4 mRNAs were differentially expressed in stromal-vascular cells (e.g., CD97 Fig 1D), and correspondingly many of these mRNAs encoded proteins expressed

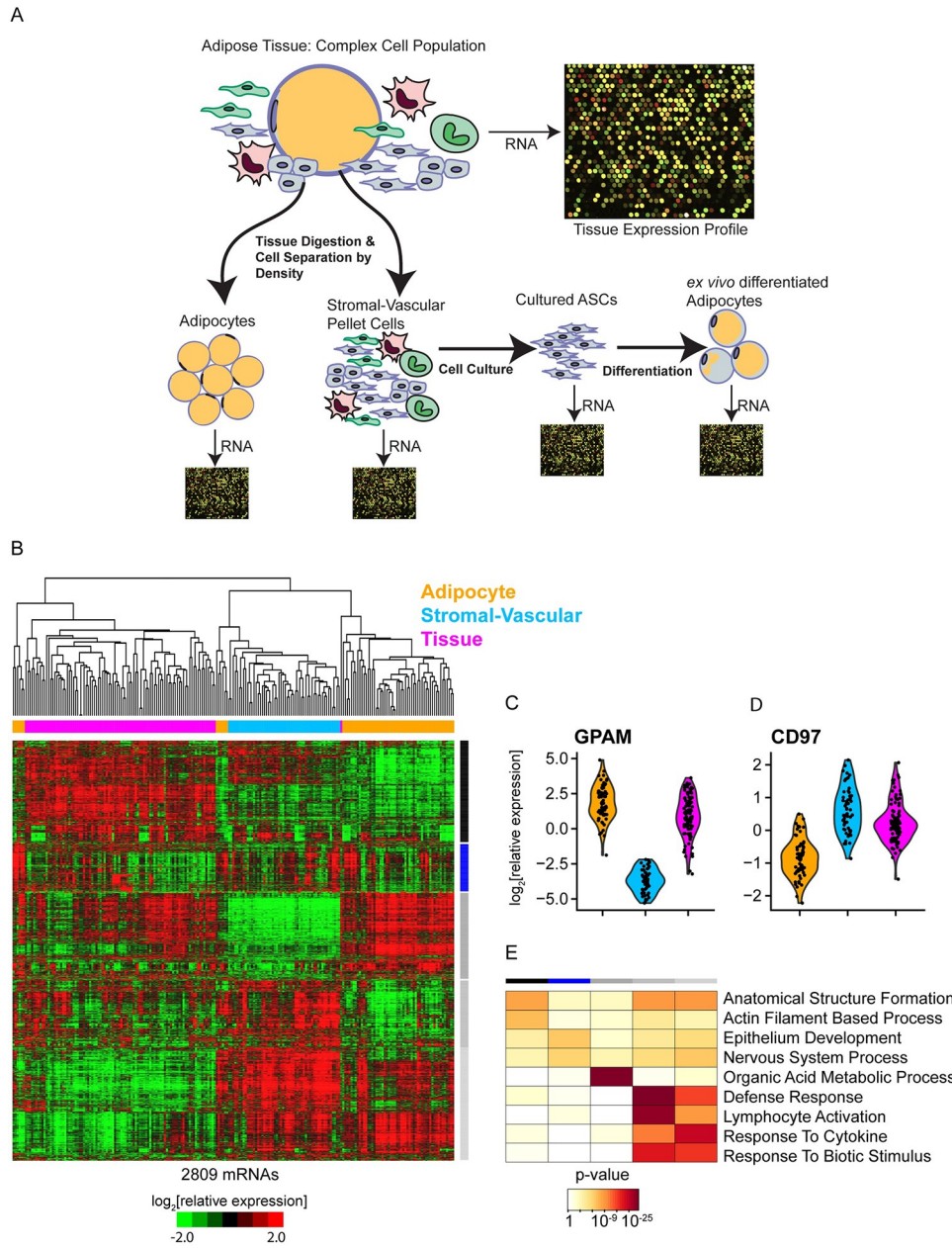

**Fig 1. Adipose tissue fractionation enriches for adipocytes in the "floating" adipocyte/lipid layer.** (A) Diagram depicting separation of complex adipose tissues into cell fractions, adipocyte and stromal-vascular cells, cultured ASCs and *ex vivo* differentiated adipocytes, for independent hybridization of RNA to DNA microarrays. (B) Unsupervised hierarchical clustering of mRNA expression across adipocytes (orange), adipose stromal-vascular cells (blue), and whole adipose tissue (magenta) from distinct anatomic depots separates samples by their fraction of origin (uncentered Pearson correlation). Variation in mRNA abundance relative to its mean across all samples is represented by a red (greater than the mean) to green (less than the mean) color scale (-1.5 to +1.5 in $\log_2$ scale). (C) Violin plot of the relative expression of adipocyte marker GPAM, which catalyzes the initial and rate limiting step in glycerolipid biosynthesis. GPAM expression is, on average, 43-fold higher in adipocytes compared to stromal-vascular cells (FDR = 0, unpaired two-class SAM). (D) Violin plot of the relative expression of CD97, which is expressed in multiple immune cell populations. CD97 expression is, on average, 6-fold higher in stromal-vascular cells compared to adipocytes (FDR = 0, unpaired two-class SAM). (E) Heatmap representation of Gene Ontology terms overrepresented among mRNAs in each of the five major clusters (indicated by bars on the right of the heatmap in Fig 1B).

in immune infiltrates and involved in "defense response" (126, $p < 1e^{-34}$). Cluster 1 included 733 mRNAs whose relative expression was highest in whole tissue, cluster 2 included 389 mRNAs with variable expression across cell fractions and cluster 5 included 715 mRNAs whose relative expression was higher in both isolated adipocytes and stromal-vascular cells than in the corresponding unfractionated adipose tissue, suggesting that these mRNAs may have been induced by the fractionation process. To further minimize the influence of non-adipocyte derived mRNAs in subsequent analyses on the mRNA expression profiles of the adipocyte fraction, we filtered out mRNAs whose relative expression in the adipocyte fraction vs. stromal-vascular fraction was in the bottom quartile among all mRNAs with quality measurements (Significance of Microarrays (SAM) unpaired t-test [42]).

## Depot of origin is a major determinant of the variation in adipocyte gene expression

We explored the relationships among expression programs of adipocytes from different depots by first ranking mRNAs by the degree to which variation in expression was accounted for by inter-depot differences, as measured by multiclass SAM. We selected the top 783 mRNAs (q-value < 0.05%) for further analyses. After removing a group of 304 coordinately expressed mRNAs potentially induced by tissue inflammation (most of these mRNAs are in cluster 2 from Fig 1B), hierarchical clustering based on expression patterns of the remaining 479 distinct mRNAs organized the samples into three major branches comprising adipocyte samples from anatomically related depots; trunk intra-abdominal depots (omental, pericolonic and retroperitoneal), trunk subcutaneous depots (abdominal and breast), and leg subcutaneous depots (thigh and lower leg), with five ectopic samples (Fig 2A, S6 Dataset). The gene expression patterns distinctly organized the sites by anatomic proximity at multiple levels; trunk vs. leg, subcutaneous vs. internal, and further, peritoneal vs. retroperitoneal, respectively. Thus, based on their gene expression programs, adipocytes from the different depots appear to be phenotypically distinct as a function of anatomic location, with anatomically distant depots differing significantly in gene expression programs, and anatomically proximal depots sharing more similar gene expression programs.

Compared to all mRNAs with reliably measured expression in adipocytes, this set selected for depot-specific variation in expression contained an abundance of mRNAs encoding proteins involved in developmental programming, including "embryo development" (61, $p < 1e^{-10}$), "skeletal system development" (45, $p < 1e^{-25}$), "tube development" (54, $p < 1e^{-8}$), "anterior posterior pattern specification" (24, $p < 1e^{-10}$) and "proximal distal pattern specification" (8, $p < 1e^{-6}$). Many of these proteins encode transcription factors (53, $p = 0.0002$). Additionally there were an abundance of mRNAs encoding proteins involved in "cell-cell signaling" (53, $p = 0.001$), "regulation of hormone levels" (25, $p = 0.0003$), "response to lipid" (34, $p = 0.0009$), "positive regulation of lipid metabolic process" (14, $p = 7e^{-5}$), "cytoskeletal protein binding" (46, $p = 0.0002$), "calcium ion binding" (27, $p = 7e^{-4}$), "signal receptor binding" (42, $p = 0.003$), "lipid binding" (28, $p = 0.02$) and "molecular transducer activity" (23, $p = 0.01$). The combinatorial expression of these transcription factors, signaling molecules, receptors and lipid metabolism proteins are potential determinants of depot-specific differentiation and functionality.

## Evidence for functional specialization among adipose depots

Most of the 479 mRNAs identified by depot-specific expression profiles were distributed among five clusters (Fig 2A), that group anatomically related sites at multiple levels (Fig 2B–2F). mRNAs whose coordinate expression patterns distinguish depots by anatomic location

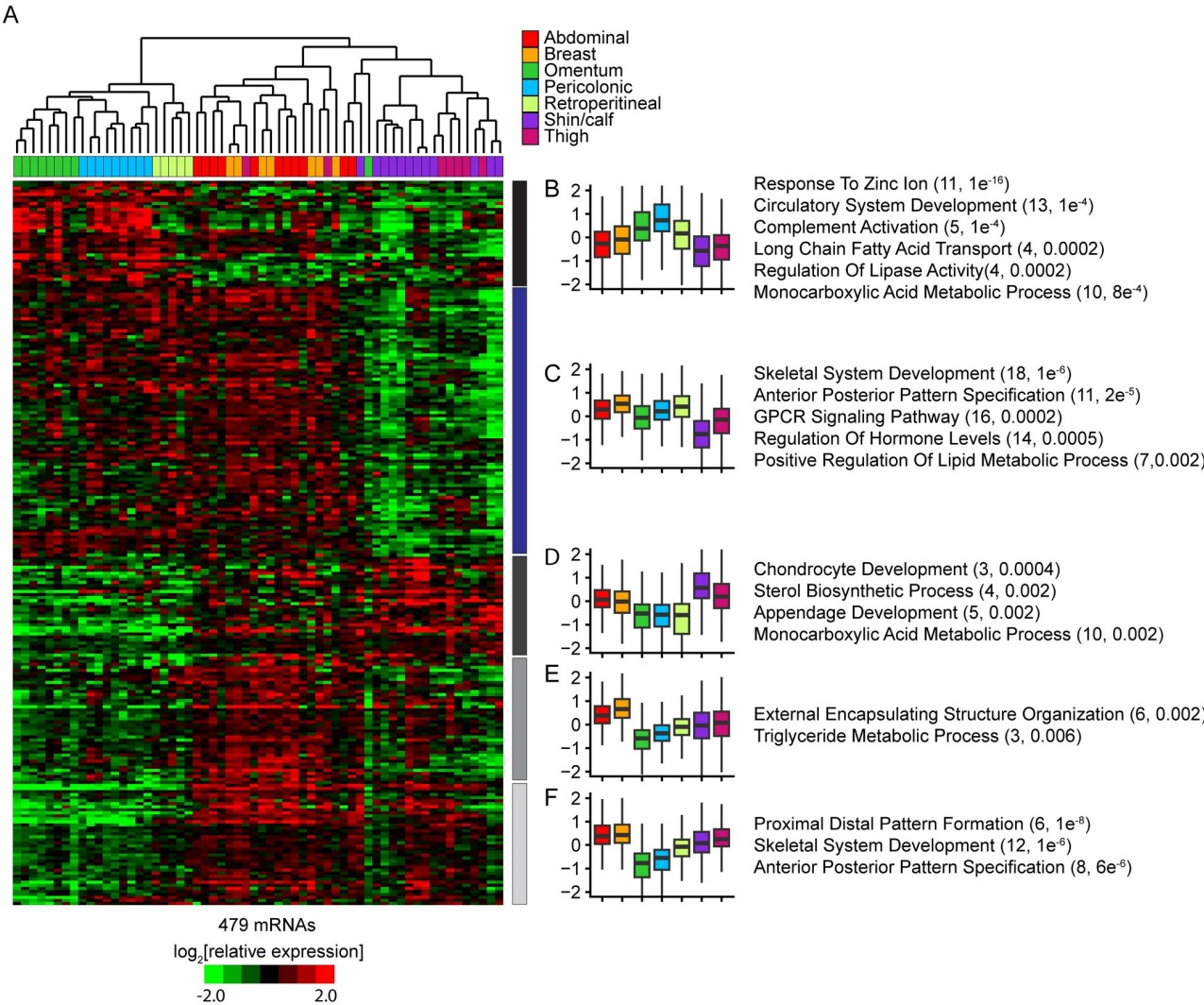

**Fig 2. Expression patterns of mRNAs that vary among adipose depots segregate sites by anatomic location.** (A) Supervised hierarchical clustering of 479 mRNAs selected for depot related differences in expression across adipocytes. Five gene clusters defined by their relative expression in adipocytes from sets of anatomically related depots are labeled to the right of the heatmap. (B) Boxplot representation of the relative expression of mRNAs in cluster 1 across depots. The bottom and top of the box represent the 1st and 3rd quartiles. The top line extends from the top of the box to the largest value no further than 1.5 times the interquartile range (IQR) from the box; the bottom line extends from the rectangle to the smallest value at most 1.5 times IQR of the box. The horizontal bar in the box is the median. To the right of the boxplot are listed representative GO terms overrepresented among these mRNAs. In parentheses are the number of mRNAs associated with the GO term and the associated Benjamini-Hochberg corrected p-value. (C-F) Boxplot representation of the relative expression of mRNAs in clusters 2–5 across depots.

may be related to physiological specialization of those depots; indeed, each of the five clusters includes genes linked to body fat distribution and adipose-related pathologies.

Cluster 2 (blue bar) includes 178 mRNAs, defined by preferential expression in trunk depots (both subcutaneous and intra-abdominal) compared to leg depots (Fig 2C). Previous studies have shown that trunk depots have higher rates of insulin-mediated glucose uptake, fatty acid uptake and lipolysis than leg depots [11, 43–45]. Trunk depots are also generally more responsive to energy surplus and weight loss than leg depots [46]. This cluster contains an abundance of mRNAs encoding proteins involved in signaling processes, including "regulation of secretion" (16, p = 0.0002), "GPCR signaling" (16, p = 0.0002),"regulation of hormone

levels" (14, p = 0.0005) and "cell-cell signaling" (27, p = 0.001). Many of these signaling processes are related to energy sensing and metabolism; for instance, cluster 2 includes the thyroid hormone nuclear receptor (THRB) (S4C Fig), angiotensin II receptor type 1 (AGTR1), corticotropin releasing hormone binding protein (CRHBP), inhibin B (INHBB) and neuropeptide Y receptor (NPY5R). Cluster 2 also includes several mRNAs encoding proteins involved in Wnt-related signaling (LRP5, SFRP1, TCF7L1, TCF7L2, and DAAM1), which is known to regulate mesenchymal stem cell fate, adipogenesis and adipocyte physiology [47]. This set also includes genes involved in systemic energy homeostasis, including insulin-mediated glucose uptake (SPTBN1, PLCL1, IGF1, FOXO4 and GRB14) and lipid metabolism (ABHD6, GDPD5, CES1, ABCA8, ABCA9, HSD17B1 and HSD17B13).

To test if the functional themes identified from this small set of mRNAs extended more broadly, we ranked all mRNAs' relative expression in trunk vs. leg samples based on SAM unpaired t-test statistic, then performed Gene Set Enrichment Analysis (GSEA) [48]. Adipocytes from trunk depots preferentially expressed many gene sets linked to energy metabolism, including "regulation of fatty acid transport" (adjusted p-value = 0.007), "oxidative phosphorylation" (p = 0.007), "positive regulation of carbohydrate metabolic process" (p = 0.008) and "fatty acid beta oxidation" (p = 0.007). Trunk depots also preferentially expressed regulators of energy metabolism, including "hormone binding" (p = 0.01), "insulin receptor binding" (p = 0.17), "integration of energy metabolism" (p = 0.01) and "signaling by GPCR" (p = 0.0002). Thus, trunk depots preferentially express genes involved in multiple levels of systemic energy homeostasis, including receptors for hormones and metabolites, regulators of glucose and lipid flux and Wnt pathway components. Preferential expression of these genes in trunk depots may be related to the differential responsiveness of trunk adipocytes to systemic regulators of adipocyte energy metabolism, including insulin and other hormones.

The 71 distinct mRNAs in cluster 1 (black bar) were preferentially expressed in intra-abdominal depots, particularly peritoneal fat (Fig 2B). Compared to the subcutaneous abdominal depot, peritoneal depots have higher rates of lipid turnover and release of non-esterified fatty acids, which are key signals of systemic energy status [11, 15]. This cluster includes mRNAs encoding proteins related to energy and nutrient sensing, signaling and metabolism. Cluster 1 includes receptors for several key regulators of lipid metabolism: the lactate receptor (HCAR1), which mediates repression of lipolysis by lactate [49], nuclear glucocorticoid receptor (NR3C1), which mediates the activation of lipolysis by glucocorticoids (see S4A Fig) [50], nuclear retinoic acid receptor (RARB), which mediates the activation of lipolysis by retinoic acids (see S4B Fig) [51], and natriuretic peptide hormone receptor (NPR1), which mediates activation of lipolysis by natriuretic peptides [52]. This cluster also includes transporters for long chain fatty acids (SLC27A2), fatty acids (SLC27A6) and glycerol (AQP9). In addition to genes involved in energy and nutrient sensing, cluster 1 includes mRNAs encoding key lipolysis enzymes (ACACB, ABHD5, ECHD).

We performed GSEA after ranking all mRNAs' relative expression in intra-abdominal depots vs. subcutaneous depots using SAM unpaired t-test scores. Adipocytes from intra-abdominal depots preferentially expressed genes involved in "lipid import into cell" (p = 0.04), "regulation of fatty acid transport" (p = 0.04), "integration of energy metabolism" (p = 0.08), "negative regulation of hormone secretion" (p = 0.04) and "GPCR activity" (p = 0.07). Thus, intra-abdominal depots, particularly the peritoneal depots, seem primed to respond to changes in energy status and then communicate those changes systematically. These energy sensing and signaling functions could be particularly important in these intra-abdominal depots given their direct access to the liver through the hepatoportal vein.

Cluster 4 (gray bar) includes 64 mRNAs preferentially expressed in trunk subcutaneous depots (Fig 2E). For most non-obese humans the abdominal subcutaneous depot is the largest

and most dynamic depot and acts as the body's primary buffer for excess energy intake [11]. Expansion and contraction of the abdominal adipose depot in response to energy status is driven mainly by adipocyte expansion (hypertrophy) and shrinkage, respectively [53]. In women, the breast adipose depot expands dramatically at puberty, temporarily contracts during lactation, and then expands during involution, primarily via hypertrophy [54]. Cluster 4 includes a number of genes with roles in regulation of cell and tissue size and shape, many of which are linked to adipose tissue expansion in obesity (ENPP2, EGFL6, RASA1, SPARC, ANXA2, SNTB2, IFT81, DIAPH2, SGCD, ATP10A, LOXL4, ATG14 and ARHGEF7). Cluster 4 also includes genes involved in sex hormone signaling (AR) and metabolism (AKR1C1-3), which regulate adipose tissue distribution, adipogenesis, and adipocyte metabolism [55].

We performed GSEA after ranking all mRNAs' relative expression in trunk subcutaneous depots vs. all other depots using SAM unpaired t-test scores. Adipocytes from trunk subcutaneous depots preferentially expressed gene sets involved in glucose flux, "cellular glycogen metabolic process" (p = 0.01), and energy storage, "energy reserve metabolic process" (p = 0.0001). Additionally, subcutaneous trunk depots preferentially expressed gene sets related to cell and tissue growth, "positive regulation of canonical Wnt signaling pathway" (p = 0.01), and cell organization and communication, such as "cell substrate junction organization" (p = 0.01), "focal adhesion assembly" (p = 0.02) and "actin mediated cell contraction" (p = 0.07). Abdominal and breast depots thus seem primed to take up and store surplus energy from glucose, and also expand and contract in response to physiological and developmental cues.

Cluster 3 (dark gray bar) includes 73 mRNAs preferentially expressed in leg depots (Fig 2D). While leg depots are generally less responsive to changes in energy status, leg adipocytes likely serve as an important long term lipid storage reservoir by taking up and storing recycled lipids rather than dietary lipids, particularly in women. Absent are the abundance of mRNAs encoding proteins linked to nutrition, energy sensing and adipose expansion seen in clusters 1, 2 and 4, which is consistent with leg adipose depots primarily functioning not as a dynamic energy reservoir, but more as a structural tissue and long term lipid storage reservoir. Cluster 3 includes a number of important adipocyte genes identified in other depots, but which may play outsized, and unappreciated, roles in leg depots. For instance, GWAS studies recently identified SNX10, a sorting nexin that regulates early endosome trafficking, as the strongest predictor of BMI-adjusted waist-to-hip ratio specifically in females [56]. SNX10 is required for adipocyte differentiation and participates in diet-induced adipose expansion specifically in female mice. CCN5/WISP2 limits adipocyte differentiation in abdominal subcutaneous adipose tissue; reduced expression is linked to adipose hypertrophy associated with metabolic syndrome, while overexpression increases insulin sensitivity and is protective against metabolic syndrome [57]. Other previously identified adipocyte genes differentially expressed in leg depots include PNPLA3/Adiponutrin, ANGPTL8, FGFBP3 and RORC/NR1F3. These results suggest that leg depots may play underappreciated roles in systemic adipose physiology, perhaps by sensing changes in energy stores and sending secreted regulatory signals.

In principle, apparently depot-specific mRNA expression could largely be driven by inter-depot variation in contaminating non-adipocyte derived cells in the adipocyte fractions. We therefore repeated the clustering analyses after selecting for mRNAs whose expression in adipocytes or ASCs was affirmed by single cell RNA sequencing [58]. Hierarchical clustering based on expression of these 352 distinct mRNAs (multi-class SAM FDR < 0.2%, 194 overlap with 479 above) yielded similar results to those described above (S2 Fig).

To test if our results extended beyond our survey, we compared our results with a recent study that identified seven human adipocyte populations based on single cell RNA sequencing of "subcutaneous" (abdominal) and "visceral" depots (omentum and peritoneum) [58].

Populations 1, 3, 4, 5 and 7 were predominantly expressed in the subcutaneous depot while 2 and 6 were predominantly visceral. We tested whether mRNAs associated with each of these seven populations were overrepresented in any of the five clusters we identified above. Indeed, mRNAs belonging to the major adipocyte population identified from abdominal subcutaneous adipocytes, "hAd1", were enriched in cluster 2 (23, $p < 1e^{-5}$, hypergeometric distribution function) and cluster 4 (18, $p < 1e^{-10}$). mRNAs in cluster 2 and 4 were preferentially expressed in trunk depots (and higher in abdominal compared to omental) and subcutaneous trunk depots respectively. mRNAs in "hAd5" were also enriched in cluster 2 (25, $p < 1e^{-8}$). In contrast, mRNAs belonging to the major visceral population, "hAd2", were enriched in cluster 1 (11, $p < 1e^{-9}$), consistent with our results.

We also examined whether the mRNAs that define these seven populations varied, as a population, across depots. As shown in S3 Fig, there is variable expression across depots, with hAd1, 3, 4, 5 and 7 more highly expressed in subcutaneous abdominal fat compared to omental, and to some degree pericolonic fat. In contrast, hAd2 was nominally higher in pericolonic and (less so) omental adipose depots as compared to abdominal depots. However, there is additional variation in expression among these groups of mRNAs beyond that accounted for by "subcutaneous" vs. "visceral" depots, highlighting the importance of investigating a wider diversity of adipocyte populations to identify additional dimensions of physiological, developmental and regulatory variation.

## PPARγ1 and PPARγ2 may have distinct targets and functions in human adipocytes

Differences between trunk and leg depots in expression of mRNAs related to glucose and lipid metabolism prompted us to explore the expression patterns of adipocyte master regulator PPARγ and its putative target genes. There are two PPARG isoforms: PPARγ2 is encoded by an alternative splice isoform and distinguished from PPARγ1 by an additional 28 amino acids at its N-terminus. The two isoforms differ in ligand-independent activity, response to and specificity for ligands, ability to activate adipogenesis, and tissue-specific expression [59–61].

The microarray used in this study included four probes for PPARG, three of which measure both PPARG1 and PPARG2 isoforms (hHC0202128, hHA035245, hHC018379) and one that specifically measures PPARG2 (hHA034694). We clustered PPARG isoforms along with a set of ~300 PPARγ targets [62, 63] based on expression in our dataset (Fig 3A, S8 Dataset). PPARG isoforms and their targets split into two main groups. The first group includes the three pan PPARG isoforms (which we will call PPARG1) plus 116 other distinct mRNAs. These mRNAs were generally more highly expressed in adipocytes from trunk depots compared to leg depots, but with considerable inter-individual variation (Fig 3B). The second cluster included PPARG2 specific isoform plus twenty other distinct mRNAs. These mRNAs were generally more highly expressed in internal trunk and breast depots compared to leg and abdominal depots, but also with considerable inter-individual variation (Fig 3C).

The PPARG1 cluster includes an abundance of mRNAs encoding proteins involved in storage of energy from glucose, including HK2, encoding hexokinase, which catalyzes the rate-limiting and first obligatory step of glycolysis, IDH3A, which catalyzes the rate limiting step in TCA cycle, ACACA, encoding acetyl-CoA carboxylase, the rate-limiting enzyme in *de novo* fatty acid synthesis, as well as other enzymes in fatty acid synthesis including FASN, DGAT1 and DGAT2. Two enzymes involved in glycogen synthesis (GYS1 and GBE1) and PPARγ coactivator CEBPA were also in the PPARG1 cluster.

The PPARG2 cluster, in contrast, comprised a set of genes encoding proteins that play major roles in lipid uptake and storage and hormone-regulated lipolysis, including LPL,

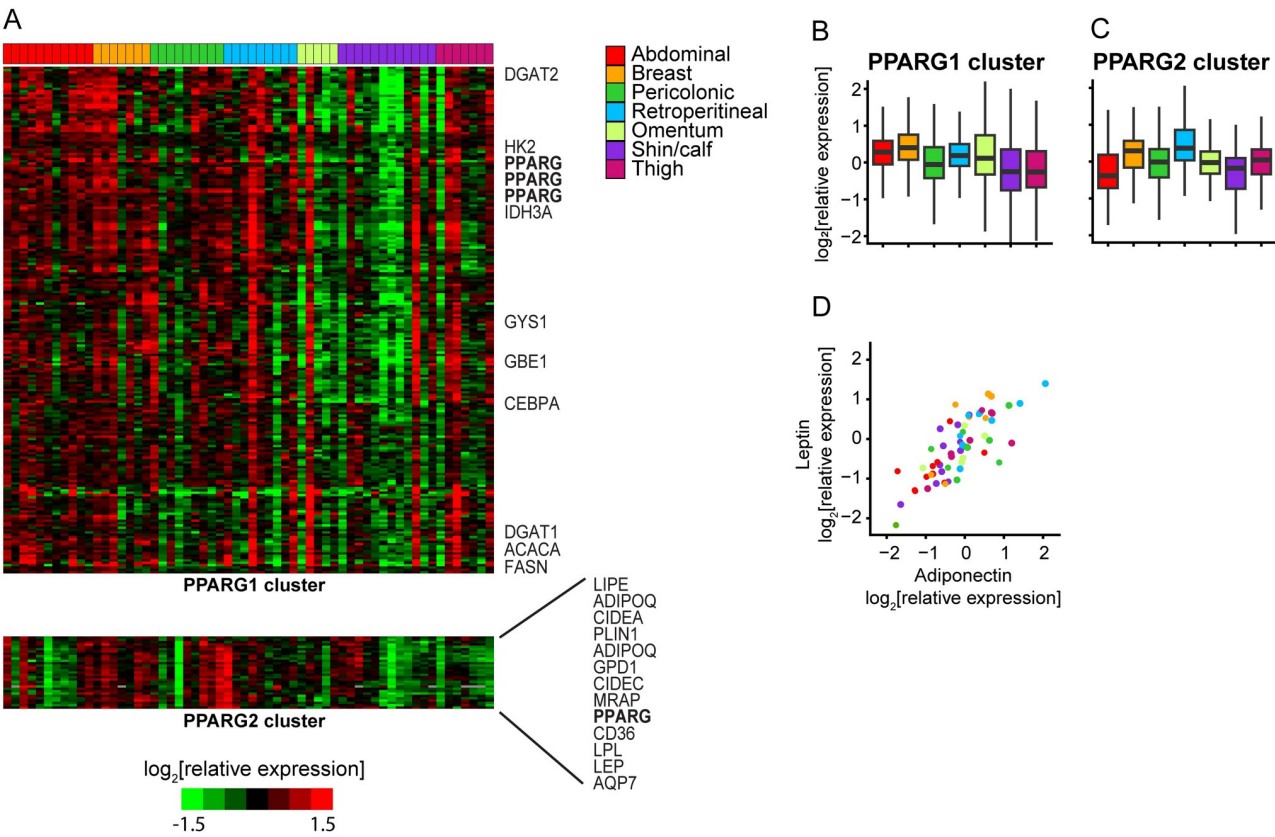

**Fig 3. PPARγ1 and PPARγ2 may have distinct targets and functions in human adipocytes.** (A) Heatmap representation of the relative expression of putative PPARγ target genes across adipocyte samples, arranged by depot. The "PPARG1 cluster" (top) includes three PPARG microarray probes and 116 other unique mRNAs. Some notable genes are listed to the right of the heatmap. The "PPARG2 cluster" (bottom) includes a probe specific to the PPARG2 isoform and 20 other unique mRNAs, twelve of which are listed to the right of the heatmap. (B) Boxplot representation of the relative expression of mRNAs in the PPARG1 cluster across depots. (C) Boxplot representation of the relative expression of mRNAs in the PPARG2 cluster across depots. (D) Scatterplot comparing the relative expression of adiponectin (ADIPOQ) and leptin (LEP) across samples. Sample depot is color-coded.

CD36, GPD1, CIDEA, CIDEC, HSL, PLIN1, AQP7, RBP7, MRAP and LIPE (Fig 3A), suggesting that these processes may be coordinately regulated in adipocytes by PPARγ2. These proteins control rate-limiting steps in lipoprotein triglyceride hydrolysis (LPL), free fatty acid import by adipocytes (CD36), and synthesis of triglycerides (GPD1), as well as lipid droplet integrity (PLIN, CIDEA and CIDEC), activation of hormone-stimulated lipolysis (PLIN, LIPE), and adipocyte glycerol release (AQP7). Coordinated expression of this set of mRNAs in adipocytes may therefore regulate their capacity for lipid uptake, storage and release. These results may provide a mechanistic explanation for why PPARG2 knock out mice have reduced adipose tissue storage rate and display metabolic inflexibility [64], which is the inability to switch between using lipid as a primary energy source in the fasting state and using carbohydrate in the fed state [65].

Genes encoding the two canonical adipokines, leptin (LEP) and adiponectin (ADIPOQ), also cluster with PPARG2. Leptin and adiponectin are almost exclusively produced by adipocytes, act on multiple target tissues and together may function as a "lipostat" to regulate food intake and energy consumption, with adiponectin acting as a signal of fasting status and leptin as signal of satiety [66]. Expression, regulation and systemic effects of these two adipokines are

reported to be in direct contrast to one another [66, 67]. In view of their opposing activities and physiological regulation, the pattern of inter-sample variation in their expression at the mRNA level was unexpected. Indeed, adiponectin mRNA ranked 19th out of ~11,000 mRNAs in correlation with leptin mRNA in this dataset (Pearson r = 0.78, Fig 3D). These results suggest that specific depots or adipocytes (differentially localized to internal trunk and breast depots) are tasked with the "lipostat" function, sensing energy stores and regulating appetite. It is surprising that within depots there is not much inter-individual variation in the ratio of LEP to ADIPOQ, at least at the mRNA level. This suggests that circulating levels of leptin and adiponectin are largely controlled post-transcriptionally, potentially via intricate feedback mechanisms involving other hormones that regulate systemic energy homeostasis, such as insulin, glucagon and FGF21 [66]. However, we also note that patients from whom these samples were obtained were most likely in a "fasting" state (pre-operatively).

In summary, PPARG1-correlated target mRNAs appear to be preferentially involved in glucose flux and *de novo* fatty acid biogenesis, while PPARG2-correlated target mRNAs encode proteins preferentially involved in lipid uptake and release, and the nutrient sensing and signaling functions of adipose tissue. Both PPARG isoforms and respective targets display extensive inter-individual variation in expression.

## HOX gene expression distinguishes adipocytes by depot of origin

In dermal fibroblasts, endothelial and smooth muscle cells isolated from various body sites and cultured *ex vivo*, the expression patterns of HOX genes distinguished cells primarily by anatomic origin, and HOX gene expression patterns may function as a kind of "positional memory" [68–71].

Hierarchical clustering of adipocyte samples based on their HOX gene expression patterns similarly grouped adipocytes by depot of origin at high resolution (Fig 4A and S5 Fig). Accordingly, a disproportionate fraction of the variation in HOX gene expression was explained by depot of origin (multi-class SAM) (Fig 4B). Each adipose depot had a distinct pattern of HOX gene expression; distinctive depot-specific patterns of the individual HOX genes yielded in aggregate a unique HOX expression signature for each depot (Fig 4A, S5 Fig and S9 Dataset). First, omental and pericolonic depots split from subcutaneous and retroperitoneal depots. Second, trunk subcutaneous depots and retroperitoneal split from leg subcutaneous depots. Third, trunk subcutaneous depots split from retroperitoneal depots. The HOX genes were exceptional (compared to randomly selected sets of 28 expressed genes) in their ability to sort adipocyte samples according to their anatomic origin (p < 1e$^{-4}$, Prediction Analysis for Microarrays (PAM)).

Likewise, comparison of HOX gene expression patterns among adipocytes from diverse adipose depots and ~150 non-adipose tissue samples from 19 organs and tissues revealed that expression patterns of HOX genes in adipocytes were generally most similar to those in nearby tissues and organs rather than to other adipocytes *per se* (Fig 4C): The pattern of HOX gene expression in omental adipocytes most closely resembled that in the stomach and lung; the HOX gene expression pattern in pericolonic adipocytes was most similar to that in the colon, uterus, and rectum; the HOX expression pattern in leg adipocytes was most like those in nerve, tendon and muscle samples from the leg; the HOX expression patterns of abdominal, breast and retroperitoneal adipocytes were most like those from breast and trunk muscle. Thus, HOX gene expression patterns appear to mark adipocytes (and non-adipose tissues) by their location in the body, rather than by cell type, consistent with a principal role in conferring positional identity.

A

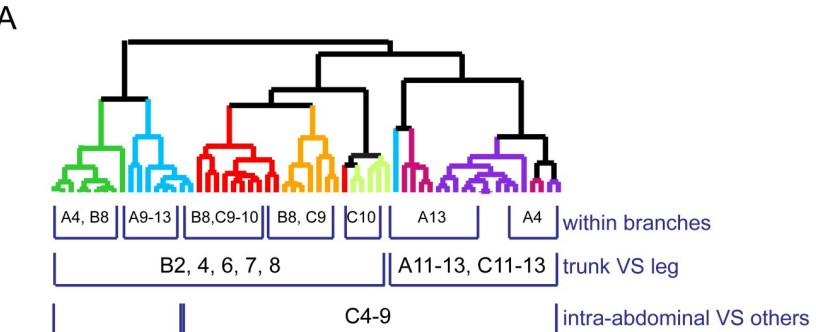

B

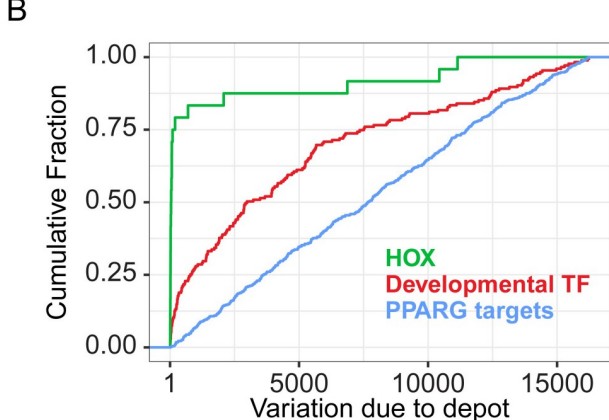

C

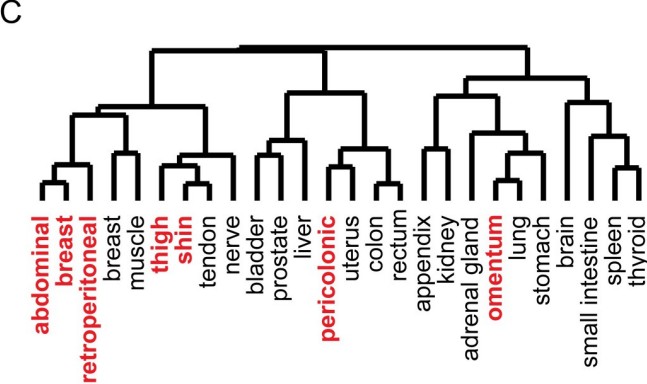

**Fig 4. HOX gene expression marks adipocytes by depot of origin and may provide intrinsic positional identity.**
(A) Dendrogram resulting from hierarchical clustering of adipocyte samples by similarity in HOX gene expression pattern shows high-resolution organization of samples by anatomic site of origin. Select genes whose expression distinguished sites, or groups of sites, are listed underneath the site or group in which they were more highly expressed. (B) Cumulative distribution of the gene rank from multi-class SAM measuring the degree to which variation in relative abundance of mRNA from a particular gene was explained by inter-depot variation (i.e., rank of 1 = gene whose variation in expression is most explained by inter-site variation). HOX genes were more likely to be highly ranked than other developmental transcription factors ($p < 1e^{-9}$), which in turn were more likely to be highly ranked than PPARγ targets ($p < 1e^{-11}$). (C) Dendrogram resulting from hierarchical clustering of adipocyte samples with non-adipose tissues based on similarity in relative HOX gene expression patterns. The median value for all samples from each adipose depot or tissue was used for each HOX gene.

## Developmental transcription factor expression profiles suggest depot-specific programming during early development in concert with anatomically related organs

We extended this analysis to look at expression of 61 transcription factors, including 18 HOX genes, chosen because variation in their expression among adipocyte samples was predominantly explained by depot of origin (FDR < 0.5%, multi-class SAM). At least 13 of these transcription factors were previously identified to have depot-specific expression in mice or humans (WT1, NR2F1, HOXA5, DMRT3, TBX15, TWIST1, DMRT2, SHOX2, HOXC8, HOXC9, IRX2, HOXC13, HOXB8) [22, 23, 25, 26]. With the exception of TBX15, our results are generally concordant with the previously published work. Most of these 61 transcription factors are implicated in developmental programming, particularly of mesodermal tissues. Several are genetically linked to variation in body fat distribution in humans (TCF7L2, TBX15, HOXC13, IRX3) [27–31, 72, 73]. Hierarchical clustering of adipocyte samples based on gene expression patterns grouped adipocytes by depot of origin at high resolution, again organizing the sites by anatomic proximity at multiple levels; trunk vs. leg, subcutaneous vs. internal, and further, peritoneal vs. retroperitoneal, respectively (Fig 5A and S10 Dataset). We interrogated the logic underlying the striking depot-specific expression profiles of these transcription factors in adult adipose depots with an emphasis on how it might connect to patterns established in early development.

Classic embryological studies in mice and pigs suggested that adipose depots are (largely) of mesoderm origin [32–35]. Mesoderm is transiently divided into three compartments during

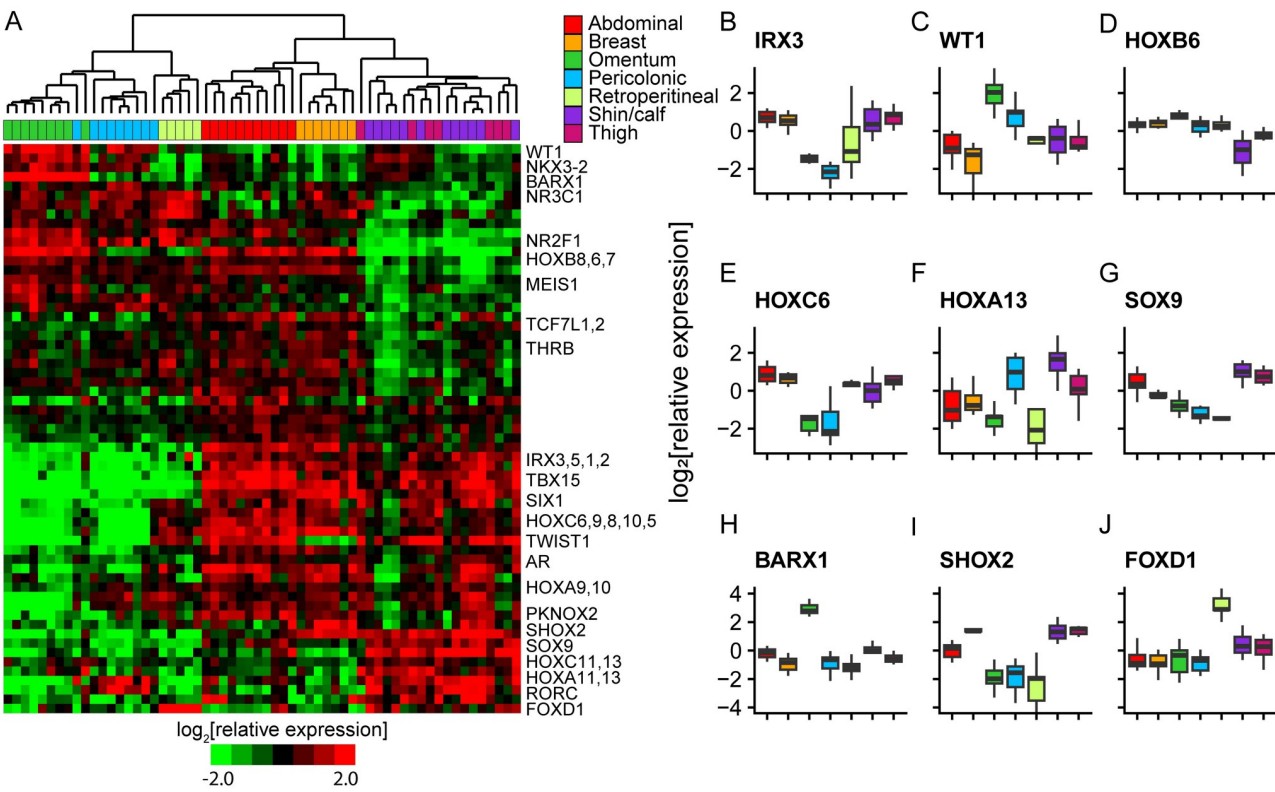

**Fig 5. Developmental transcription factor expression profiles suggest depot-specific programming during early development.** (A) Supervised hierarchical clustering of mRNAs encoding transcription factors selected for depot related differences in expression across adipocytes (FDR < 0.5%, multi-class SAM). (B-J) Boxplot representation of the relative expression of example developmental transcription factors across depots.

early development: paraxial, intermediate and lateral plate mesoderm [74, 75]. Lateral plate mesoderm splits into splanchnic and somatic pouches. Splanchnic mesoderm gives rise to the cardiovascular system and the mesentery, while somatic mesoderm forms on the inner surface of ectoderm and is the source of appendicular skeleton and ventral dermis [76]. Lineage tracing of "marker" genes in mice suggest that visceral adipocyte depots derive from splanchnic lateral plate mesoderm (WT1[+], HOXB6[+]) while subcutaneous depots derive from somatic lateral plate mesoderm (PRRX1[+], HOXB6[+]) [21, 36, 37, 39]. In contrast, retroperitoneal and dorsal depots are believed to derive from paraxial mesoderm (MEOX1[+], HOXB6[-]) [21, 37].

We find that adult human subcutaneous adipose depots specifically expressed IRX1, IRX2, IRX3, IRX5, TBX15, TWIST1, SETB2, AR, ZFHX4, HOXC5, HOXC6, HOXC8 and HOXC9 (Fig 5A and 5B). Several of these factors are implicated in somatic lateral plate mesoderm developmental programs (IRX3, IRX5, TWIST1, TBX15) [77–81]. Genetic variants in an IRX3 enhancer (FTO locus) are reproducibly associated with obesity and type 2 diabetes, though depot-specific roles for IRX3 have not been explored [30, 31]. TBX15 genetic variants are recurrently linked to waist to hip ratio [27, 28]. More severe mutations in TBX15 are linked to Cousin syndrome and defects in craniofacial development. These results are congruent with lineage marker experiments in mice that suggest subcutaneous depots derive from somatic lateral plate mesoderm [38, 39]. Beyond that, our results implicate over a dozen developmental transcription factors whose differential expression in subcutaneous adipocytes likely imparts depot-specific identity during development, in concert with adjacent superficial tissues, and maintains their identity into adulthood.

Peritoneal adipose depots specifically expressed WT1, NKX3-2/BAPX1 and ISL2 (Fig 5A and 5C). WT1 is implicated in kidney, gonad and heart development, and is considered a marker of splanchnic lateral plate mesoderm [76]. In mice, visceral adipocytes derive from WT1[+] progenitor cells [36]. NKX3-2/BAPX1 is also an early marker for splanchnic lateral plate mesoderm in mice and is implicated in spleen and pancreas development [82–84]. Thus, our results are consistent with adipocytes from peritoneal depots originating from splanchnic lateral plate mesoderm.

In this study, adipocytes were distinguished along the anterior-posterior axis by combinatorial expression of homeobox genes. Most notable are the differences between trunk and leg depots. All five trunk depots preferentially expressed HOXB2, HOXB4, HOXB6, HOXB7, HOXA4 and HOXA5 (Fig 5A). HOXB6 marks lateral plate mesoderm during gastrulation [37, 85], but expression appeared to be lost in the most posterior (lower leg) mesoderm derived cells in adulthood, based on this study and published work on *ex vivo* cultured fibroblasts [70] (Fig 5D). Subcutaneous and retroperitoneal depots preferentially expressed SIX1 and HOXC5-10 (Fig 5E), while subcutaneous leg depots preferentially expressed posterior HOX genes, HOXA11, HOXA13 (Fig 5F), HOXC11 and HOXC13, as well as SOX9 (Fig 5G), ZIC1 and RORC. SOX9 is implicated in cartilage and chondrocyte development and could potentially play a role in specifying the more structural roles of these depots [86].

Adipocytes from anatomically related depots were further distinguished from each other by combinatorial expression of developmental transcription factors. For instance, stomach mesenchyme master regulator BARX1 was specifically expressed in omental adipose cells (Fig 5H), and HOXB8 was preferentially expressed in omental compared to pericolonic adipocytes. In contrast, HOXA11 and HOXA13 (Fig 5F) were preferentially expressed in pericolonic compared to omental. Adipocytes from breast and abdominal depots were distinguished from each other by the preferential expression of PKNOX2 and SHOX2 (Fig 5I) in breast and HOXC10 in abdominal.

In mice, retroperitoneal adipocytes derive from paraxial mesoderm during gastrulation, while the bulk of the kidney and other urinary tract organs arise from intermediate mesoderm,

which has not been demonstrated to produce adipocytes [37, 87]. The kidney is surrounded by the renal capsule, a stromal structure that plays a key role in kidney organogenesis. The retro-peritoneal adipose depot surrounds the renal capsule. The metanephric mesenchyme, including the renal capsule, derives from FOXD1$^+$ progenitors during organogenesis, which are located at the outer surface of the developing kidney [88, 89]. We found that retroperitoneal adipocytes specifically expressed FOXD1 (Fig 5J), suggesting that this master regulator may help specify the retroperitoneal adipose depot as part of kidney development. Neither splanchnic nor somatic lateral plate mesoderm markers were expressed at high levels in retroperitoneal adipocytes, consistent with a non-lateral plate origin of this depot. Retroperitoneal adipocytes share some physiological (Fig 2) and developmental expression signatures with both the internal and subcutaneous trunk adipocytes; for instance, LMO3 and RARB, which are both implicated in "visceral" adipocyte metabolic control [90, 91], are most highly expressed in retroperitoneal adipocytes, with intermediate expression in peritoneal depots, while HOXC5-10 are most highly expressed in retroperitoneal, abdominal and breast adipocytes.

In sum, our results suggest combinatorial control of position-specific developmental specialization of adipose depots during gastrulation and embryogenesis, in concert with anatomically related organs and tissues, whose unique identity and functional specialization are more easily recognized.

## Maintenance of site-specific differences *ex vivo*

The observed differences in expression of hundreds of mRNAs among adipocytes isolated from different depots could represent physiological responses dictated by their distinct tissue microenvironments or they could be developmentally programmed differences that specify the differentiation of anatomically distinct fat depots into functionally distinct adipose "organs", just as they define more conventionally defined organs with distinct functions at distinct anatomical sites. One characteristic of the distinct differentiated cells that define conventional organs is that they retain recognizable characteristics of their unique gene expression programs even when separated from their native microenvironment and cultured *ex vivo* [92]. We therefore examined whether site-specific differences in gene expression observed between adipose cells from different depots would be preserved when they were cultured outside of their native anatomical context and local microenvironment. Does expression of developmental transcription factors and other differentially expressed genes in cultured adipose stem cells and adipocytes differentiated *ex vivo* mirror that in adipocytes directly isolated from the same depot?

To address these questions we obtained adipose stem cell (ASC) samples from five of the seven distinct adipose depots (abdominal, breast, omentum, shin/calf, thigh) (see Materials and methods). We cultured the cells for several passages and induced adipocyte differentiation for a portion of the cells. Total RNA was isolated from 46 cultured ASCs and 42 *ex vivo* differentiated (EVD) adipocytes and analyzed by DNA microarray hybridization. The *ex vivo* differentiation procedure yielded adipocytes from each ASC sample, reflected in the development of visible lipid droplets as well as changes in gene expression, including robust induction of hundreds of mRNA targets of adipogenic transcription factor target PPARγ (S6 Fig) and many mRNAs encoding proteins linked to metabolic programs associated with mature adipocyte function. Specific biological processes represented among the mRNAs genes whose expression increased during differentiation (690 mRNAs with >3 fold-change on average and FDR < 0.1%) are highly enriched for annotations related to lipid metabolism such as "lipid

metabolic process" (154, p < 1e$^{-38}$), "lipid biosynthetic process" (83, p < 1e$^{-19}$) and "lipid catabolic process" (53, p < 1e$^{-20}$).

While each culture appeared to produce *bona fide* adipocytes, the degree to which each culture differentiated varied from sample to sample, regardless of site of origin (S6 Fig). Expression levels of PPARγ target genes, and potentially the extent of differentiation, were significantly lower in omentum samples compared to the subcutaneous samples, consistent with previous publications [93, 94]. To measure the influence of differentiation on variation in mRNA expression across EVD adipocytes, we extracted principal components of EVD adipocyte expression data and compared them to the average fold-induction of PPARγ targets during differentiation for each sample. PPARγ target induction significantly correlated with the first principal component (Pearson r = 0.96), which explained 24% of the variance in the dataset; heterogeneity in the extent of differentiation was thus likely a major source of variation in expression among EVD adipocytes regardless of depot of origin.

## Depot-specific patterns of developmental transcription factor expression are retained *ex vivo*

We tested the prediction that a characteristic pattern of developmental transcription factor expression is an intrinsic feature of adipose precursors, by tracing expression of 54 transcription factors identified as having site-specific expression *in vivo* (Fig 5A), for which we had good data in cultured adipose cells. We found that the expression patterns of HOX genes and other developmental transcription factors in cultured ASCs and EVD adipocytes closely matched the depot-specific patterns we observed in the adipocytes isolated directly from the corresponding fat depots (Fig 6A). Indeed, for 32 of these genes, the Pearson correlation of the median depot-specific expression values across five depots for *in vivo* adipocytes and adipose cells cultured *ex vivo* (ASCs and EVD adipocytes) was greater than 0.7 (vs. only 1/54 expected based on the correlations for all well measured mRNAs, p < 1e$^{-37}$). These results support the idea that the depot-specific patterns of expression of these developmental transcription factors in mature adipocytes reflects intrinsic differentiation programs that are established in local adipocyte precursors during embryogenesis and persist through differentiation to mature adipocytes, even in the absence of continuing depot-specific extrinsic signals.

## Gene expression programs reflecting functional specialization of adipose depots are maintained by adipocytes differentiated *ex vivo*

The evidence for site-specific developmental programming of adipose cells suggests a possible mechanism for intrinsic maintenance of the gene expression patterns that underlie the distinct functional specialization of adipocytes in different depots. We sought to determine to what extent the observed differences in global gene expression programs from different depots were maintained in our *ex vivo* differentiated adipocytes. We tested whether mRNAs that distinguish pairs of sites in freshly isolated adipocytes (200 most significant mRNAs in each direction) were similarly differentially expressed between the corresponding *ex vivo* differentiated adipocytes, as measured by the Pearson correlation between the mRNAs' average difference in relative expression between the samples. For example, the scatter plot in Fig 6B shows the results for the comparison of omentum vs. shin adipocytes (Pearson r = 0.58). For all 10 pairwise comparisons, the correlation was positive, and highly significant, ranging from 0.31 to 0.58 (Fig 6C). Therefore, adipocyte precursors differentiated *ex vivo* retain depot-specific mRNA expression profiles, suggesting many of these differences are intrinsically programmed.

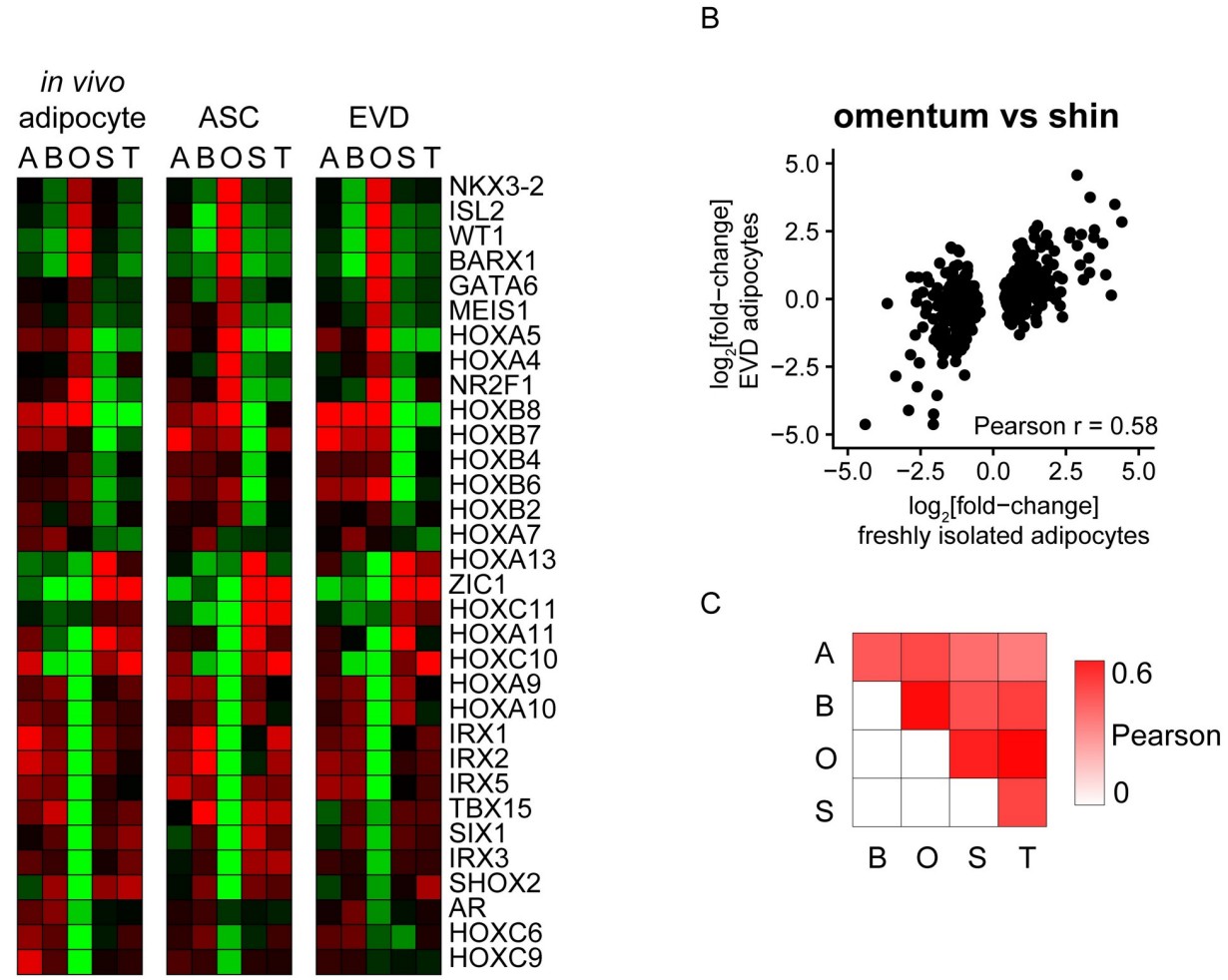

**Fig 6. Depot specific expression profiles are retained *ex vivo*.** (A) Heatmap representation of the relative expression of a set of 32 developmental transcription factors (from the set of 54 with quality data) for which the Pearson correlation of the median depot-specific expression values across five depots for *in vivo* adipocytes and adipose cells cultured *ex vivo* (ASCs and EVD adipocytes) was greater than 0.7. Sites are ordered left to right: A = Abdominal, B = Breast, O = Omentum, S = Shin/Calf, T = Thigh. (B) Scatterplot comparing the average expression differences between omentum and shin/calf depots in freshly isolated adipocytes (x-axis) and EVD adipocytes (y-axis). We selected the 200 mRNAs most significantly different in each direction in freshly isolated adipocytes (SAM t-test). (C) Heatmap representation of the Pearson correlations for each pairwise comparison described in (B).

## Independent validation of our results

In this study, we have drawn conclusions about adipocyte physiological and developmental specialization based on a relatively small sample size, while not explicitly accounting for potential confounding factors, using a single gene expression platform. We therefore used two approaches to cross-validate our results.

First we asked if our gene expression measurements could be reproduced using reverse transcription-quantitative polymerase chain reaction (RT-qPCR). We performed Taqman RT-qPCR assays on 50 mRNAs in 51 of 59 samples profiled by DNA microarray, plus twenty-six new samples (S11 Dataset). We obtained quality RT-qPCR measurements for twelve developmental transcription factors, and for eleven of them (TBX15, SOX9, SHOX2, HOXC8, HOXA13, HOXB7, NKX3-2, TWIST1, ZIX1, FOXD1 and HOXB8) the microarray and RT-

qPCR data are highly correlated and show the same depot-specific patterns of expression (S7 Fig and S2 Table). We obtained RT-qPCR data on at least 14 putative PPARγ targets (PPARG1, ACACA, ACACB, FASN, PFKFB3, GYS1, PPARG2, LIPE, CIDEC, PLIN1, GPD1, CD36, LPL and AQP7), which affirm that these genes display extensive inter-individual variation in expression regardless of depot (S8 and S9 Figs and S2 Table). LEP and ADIPOQ showed a modest correlation in expression based on RT-qPCR (Pearson r = 0.34) (S10 Fig). Thus, the RT-qPCR results generally support the microarray results.

The samples we studied were incidental to procedures associated with specific medical or physiological factors (i.e., cancer, diabetes, former obesity, S1 Table) and biased towards female donors in several depots. Variations in the underlying conditions, procedures and sex of donors could confound interpretation of apparently depot-specific differences in gene expression.

The GTEx project includes genomic profiles of thousands of samples collected from 54 non-diseased tissue sites, including unfractionated adipose tissue from omentum and "subcutaneous leg" [95]. To see if our results extended to this dataset, we focused on the 479 genes whose variation in expression was largely driven by depot of origin (Fig 2). If indeed depot of origin is the primary driver of variation in expression of these genes, we expect to see similar differences in expression levels when comparing omentum and leg adipocytes in our study vs omentum and leg adipose tissue in GTEx, despite differences in donor genetics, pathophysiology, tissue preparation and expression platform. As shown in Fig 7A and 7B, the concordance between our study and GTEx was excellent (Pearson r = 0.78), providing independent validation of our results.

There is extensive sexual dimorphism in adipose distribution and response to energy flux. We did not account for the sex of sample donors in our study. Since the GTEx collection includes samples from hundreds of male and female donors, we compared the relative expression of these 400+ genes between female and male donors in each site. For both sites, average expression for these 400+ genes was strikingly concordant between female and male donors (Fig 7C), with much more variation in expression accounted for by depot (Fig 7B). These analyses demonstrate that our results extend to a much larger independent dataset and are not driven by donor sex, pathophysiology or expression platform.

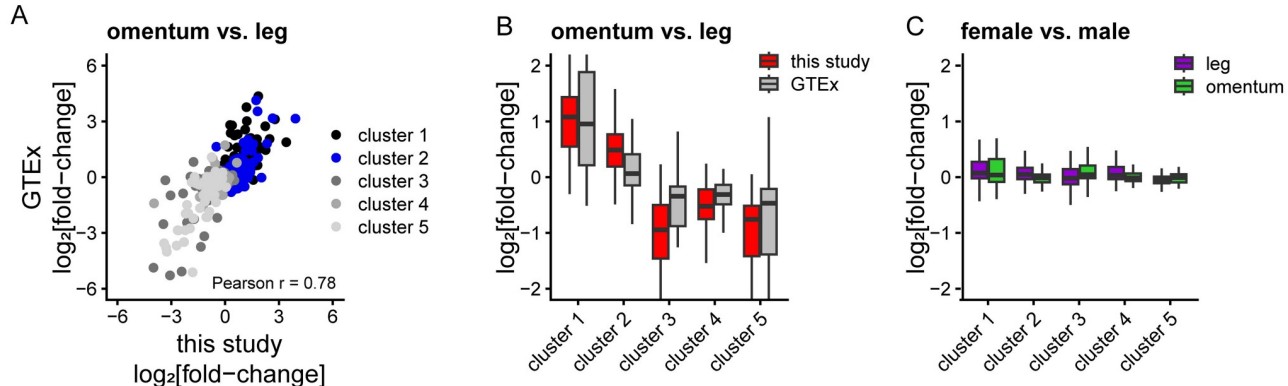

**Fig 7. Depot-specific expression patterns identified herein are supported by GTEx RNA sequencing data from omentum and leg adipose tissue.** (A) Scatterplot comparing the average expression difference of ~400 genes from Fig 2 between omentum and leg adipocytes from this study (x-axis) and omentum and leg adipose tissue from GTEx collection (y-axis). Genes are color coded by cluster according to Fig 2. (B) Boxplots of expression differences of the genes from clusters 1–5 in Fig 2 between omentum and leg for this study (red) and GTEx collection (grey). (C) Boxplots of expression differences of the genes from clusters 1–5 in Fig 2 between female and male donors from the GTEx collection in leg (purple) and omentum (green).

## Discussion

We examined global gene expression patterns in 83 adipose tissue specimens from 13 different depots; for 60 of these specimens, representing seven depots, we also profiled expression in isolated adipocyte and stromal-vascular cells, and in some cases in cultured adipose precursor cells both before and following *ex vivo* adipocyte differentiation.

The experimental results and analyses reported here provide compelling evidence that adipose depots at different anatomic sites are distinct, developmentally and physiologically specialized organs. Moreover, they reveal multidimensional molecular variation in adipocytes, with diverse and extensive variation among depots as well as among individuals.

### Depot-specific functional specialization

Gene expression profiles of adipocytes from various depots show extensive variation in expression of hundreds of mRNAs across sites. Clustering analyses tended to organize the samples by depot of origin, and to group anatomically proximal sites, demonstrating that site of origin substantially influences adipocyte global gene expression programs and generally dominated over other potential sources of variation such as pathophysiology, sex, or age.

Previous studies have revealed differences between the omental, abdominal and upper leg fat depots in specific aspects of metabolism, such as rates of glucose uptake and lipolysis [12–20], but exploration of the extent and diversity of depot-specific metabolic specialization has limited. We discovered that, compared to leg depots, adipocytes from trunk depots differentially expressed genes involved in canonical adipocyte metabolism and nutrient/metabolite signaling processes that integrate metabolic programs. We also observed subtle, but likely important, differences between the peritoneal and subcutaneous trunk depots in expression programs related to adipocyte metabolism and signaling systems. These may reflect intrinsic differences in the physiological roles of these distinct depots. The peritoneal depots had higher expression of key genes involved in nutrient signaling (e.g., lactate, glycerol, fatty acids, glucocorticoids, retinoic acid) and lipid mobilization. These depots may have a relatively prominent nutrient sensing function, which, especially if coupled to release of signaling molecules, could be particularly important in these depots given their direct access to the liver through the hepatoportal vein. The subcutaneous trunk depots differentially expressed genes involved in carbohydrate metabolism and in regulating cell structure. In most individuals the abdominal subcutaneous fat depot is the primary site of expansion during states of positive energy balance. Since expansion of abdominal subcutaneous adipose tissue is relatively unconstrained by physical limits, it is well suited for collection and storage of energy from excess carbohydrate.

Putative targets of PPARγ1 and PPARγ2 displayed extensive inter-individual variation in expression, regardless of depot, highlighting their responsiveness to physiological cues. Despite extensive inter-individual variation, these genes, on average, were expressed at lower levels in adipocytes from the thigh and lower leg depots, suggesting that insulin-stimulated glucose uptake may be a less important function in these depots. The idea that adipose depots in the leg may be relatively inert with respect to lipogenesis, lipid uptake and turnover is consistent with the observation that the fat depots in the lower leg are relatively resistant to expansion during weight gain, though there is considerable inter-individual variation in this propensity. Leg depots differentially expressed several genes that have previously only been studied in trunk depots. Several of these genes encode secreted proteins with signaling functions, suggesting that leg depots may play underappreciated roles in regulation of systemic energy metabolism.

## Depot-specific developmental programming

There is limited information regarding the embryologic origin and development of various adipose depots, particularly in humans. Our results suggest answers to critical questions regarding the developmental origins of adipose depots. How and when are depot-specific characteristics programmed? Do they persist in the face of differences in physiology or when divorced from the *in vivo* microenvironment, preserving a memory of adipocyte "positional and developmental identity" [68–71]? What is the developmental context for depot-specific programming of adipocytes?

HOX gene expression patterns were uniquely depot-specific and tended to resemble those in nearby organs and tissues, making these genes strong candidates for determinants of adipocyte positional identity. The patterns of expression of the HOX genes and other developmental transcription factors provide support for recently proposed developmental origins of specific depots, based on lineage tracing experiments in mouse: adipose tissues from the subcutaneous depots, like other superficial mesenchymal tissues, arise from somatic lateral plate while adipose tissues from intra-abdominal depots derive from splanchnic lateral plate mesoderm [21, 36, 37, 39]. Expression patterns of other transcription factors with major roles in early developmental programs also suggested a general link between development of non-adipose tissues and adjacent adipose depots; differences in expression of these factors may be a result of anatomic location during development. Site-specific expression of these developmental transcription factors genes was exceptionally preserved in ASCs and EVD adipocytes even following prolonged separation of adipocytes from their native microenvironment, which is consistent with epigenetic "hardwiring" of these genes during development and with a role in preserving depot "identity" through adulthood [68–71].

The mechanisms by which depot-specific expression of developmental regulatory genes might give rise to and maintain functionally specialized adipose tissues are of great interest. The simplest possibility is that these transcription factors combinatorially tune the depot-specific expression of genes with important adipose functions. Natural genetic variation and loss of function studies provide compelling support for this hypothesis; NR3C1, RARB, LMO3, FOXO4, TEAD1, FOXD1, AR, THRB, RORC, TBX15, TWIST1, TCF7L2, SHOX2, HOXC13, IRX3 and IRX5 are each phenotypically linked to adipose tissue functions, body fat distribution, obesity or type 2 diabetes [24, 27, 29, 96–99]. Our work identifies dozens of site-specific transcription factors that can now be examined for their roles in dictating site depot-specific phenotypes.

## Medical perspective

Anatomic depot-specific differences in response to positive energy balance are deeply intertwined with susceptibility to obesity-associated diseases, including type 2 diabetes, cardiovascular disease and metabolic syndrome [7, 8].

Many of the depot-specific developmental transcription factors and other regulatory proteins identified here are genetically linked to obesity, variations in body fat distribution or adipose tissue function. Genetic variation affecting these regulatory factors may alter developmental programming and thereby adipocyte metabolism, communication with other cell types, or expansion capacity. For example, some variants may differentially promote expansion of a specific depot, correspondingly altering body fat distribution, or alter depot-specific physiological, metabolic or endocrine functions, with resulting parallel effects on global energy metabolism and depot size. Genetic variants in an enhancer for IRX3, a developmental marker of somatic lateral plate mesoderm, are recurrently linked with body fat distribution and obesity [30, 31]. We found that IRX3 expression was highly specific to

subcutaneous adipose depots, suggesting a role in development and maintenance of subcutaneous depot-specific phenotypes. Genetic variants of TCF7L2 are strongly linked to development of type 2 diabetes [72, 73, 100]. Mice with adipocyte specific knockout of TCF7L2 in adipocytes had impaired glucose tolerance and insulin sensitivity and increased adipose tissue mass and body mass, which was accompanied by adipocyte hypertrophy in subcutaneous adipose tissue [101]. We found that TCF7L2 is preferentially expressed in trunk subcutaneous depots and hypothesize that misregulation of TCF7L2 in abdominal adipocytes could contribute to its reported links to type 2 diabetes.

The data and analyses presented here provide new insights and new clues for further investigation into human adipocyte diversity, site-specific developmental programming, and the unique contribution of various depots to different adipose tissue functions. Future studies of the mechanisms and role of adipose depot-specific gene expression could provide a route to a better mechanistic understanding of the association between body fat distribution, genotype, diet and metabolic diseases, and move us closer to individualized management of obesity and associated diseases.

## Materials and methods

### Specimen collection

Adipose tissue specimens were collected either from the Stanford Tissue Bank, Stanford Dermatology Clinic, Stanford Surgical Pathology or directly from the surgeon at the time of surgery. All surgeries were performed at Stanford Hospital and Clinics. When necessary, a trained pathologist dissected the desired adipose specimen from the organ/limb/skin to which it was attached. Samples acquired by Stanford Tissue Bank or Surgical Pathology may have been stored at 4°C for up to several hours but were always processed on the same day as the surgery was performed. All samples are considered "normal human tissues" and were originally collected under the IRB "Study of Messenger RNA Expression in Normal Human Tissues" ID: 76363, which did not require patient consent as very little patient information (age, gender, procedure) was provided. Samples were collected between September 1, 2002 and December 31, 2008.

### Tissue fractionation

Adipose tissue samples were thoroughly rinsed with 70% ethanol to minimize risk of microbial contamination of cell cultures. Tissues were then rinsed with Hanks Balanced Salt Solution (HBSS) and transferred to a clean dish. At this time 1–4 g of tissue were removed and stored at -80°C. Additionally, a thin slice of each tissue was placed in 10% formalin for paraffin embedding. The remainder of each tissue sample was minced using surgical scissors. Tissue pieces were transferred to 50 ml conical tubes (up to 25 ml tissue per tube), and an equivalent volume of collagenase solution* was added to the tube. Tissue was again minced with scissors until no large pieces remained. Tubes were shaken to mix and incubated at 37°C for 30 min to 4 hours, until pieces of tissue were no longer visible. Shin/calf adipose tissue in general took the longest and abdominal and thigh lipoaspirate the shortest time to digest. Digested tissues were then passed through a 180-micron nylon net filter and centrifuged at 100g for 5 minutes to pellet stromal-vascular cells. A 16-gauge needle attached to a 20 ml syringe was passed through the top visible lipid/adipocyte layer and the entire bottom layer containing buffer, cell pellet and blood cells was aspirated from the bottom and transferred to a new conical tube. This cell suspension (stromal-vascular fraction) was thoroughly mixed by pipetting and divided equally into two conical tubes. One of these was set aside for cell culture (see "Cell Culture") and the other was centrifuged at 400g for seven minutes. Buffer was carefully aspirated with a pipette

to avoid losing pelleted cells and 15 ml Trizol was added. The remaining lipid/adipocyte layer was washed 1–2 times depending on the amount of visible blood cells: 25 ml HBSS was added to the adipocytes, and gently inverted to mix. Adipocytes were centrifuged at 100g and excess buffer along with contaminating cells was aspirated from the bottom using a needle and syringe as described above. An equal volume of Trizol was added to the washed adipocytes. The stromal-vascular pellet sample and the adipocyte sample were stored in Trizol at -80˚C.

*Collagenase Solution: For stock solution, 1000 U/ml Sigma Type II Collagenase (Cat# C6885) was dissolved in HBSS and centrifuged at 1000g for 20 minutes. The resulting liquid fraction was filtered (0.22 micron) and stored at -20˚C and pelleted crystals were discarded. For working solution, 1000 U/ml collagenase stock solution was diluted 3x in HBSS containing 5 mM $CaCl_2$.

## Cell culture

One-half of each stromal-vascular pellet fraction from the "Tissue Fractionation" procedure was reserved to culture adipose stem cells. The cell suspension was centrifuged at 400g for seven minutes and the buffer was carefully aspirated with a pipette to avoid losing pelleted cells. Cells originating from the pericolonic or omental depots were additionally treated for ten minutes with RBC lysis buffer*, again centrifuged, and the lysis buffer aspirated. Cells were resuspended in 10 ml DMEM + 10% FBS + 2.5 μg/ml Fungizone (Invitrogen Cat# 15290–018) + Pen/Strep/Glutamine (Invitrogen 10378–016). Cells were transferred to the appropriate size culture flask (10–20 g starting tissue in one T25 flask, 20–50 g in one T75 flask). Additional DMEM (~1 ml/5 $cm^2$) was added to cells and mixed by pipetting. Cells were incubated overnight or up to 24 hours to allow time for adipose stem cells to adhere. Cells were rinsed with warm PBS and given fresh medium. Presence of adherent stromal cells was assessed visually at 10X magnification. Successful cultures were continually grown in DMEM + 10% FBS + Pen/Strep/Glutamine. Fungizone was no longer added after two passages. Cells were split 1 to 3 at 70–90% confluence. When enough cells were obtained (at least one confluent T225 flask), cells were rinsed with PBS and trypsinized, pelleted by centrifugation and resuspended in fresh medium. Each of 2, 10 cm dishes were plated, one for pre- and one for post-differentiation RNA collection. For the latter see "Adipocyte Differentiation". Cells for pre-differentiation RNA collection were plated at 100% confluence in fresh DMEM and incubated overnight. Cells were photographed at 10X magnification to document confluence and morphology of cells. DMEM was aspirated and cells were rinsed with warm PBS. PBS was replaced by six ml Trizol. Plates were gently scraped and Trizol/cell lysate was transferred to a 15 ml conical tube and stored at -80˚C.

*RBC lysis buffer: In 1 liter sterile water, 8.26 g ammonium chloride ($NH_4Cl$), 1 g potassium bicarbonate ($KHCO_3$), and 0.037 g EDTA.

## Adipocyte differentiation

Adipose stem cells were grown to confluence as described in "Cell Culture". Cells were rinsed with warm PBS and trypsinized. Trypsinized cells were transferred to a 15 ml conical tube, pelleted by centrifugation and resuspended in 5 ml DMEM + 10% FBS + Pen/Strep/Glutamine (Invitrogen Cat# 10378–016). Cells were plated to achieve the greatest density possible depending on the number of cells available, as higher confluence yielded a greater percentage of differentiating cells. In general, cells were plated at 100–300% confluence. Cells were incubated overnight and photographed the following day at 10X magnification to document confluence and record the morphology of cells. Cells were rinsed with warm PBS. PBS was replaced with Zen-Bio differentiation medium (Cat# DM-2) and the differentiation procedure

was followed as described in Zen-Bio instruction manual ZBM-1. At 16 days following induction of differentiation, cells were photographed at 10X magnification to document the number and morphology of differentiating cells. Cells were rinsed with warm PBS. PBS was replaced with six ml Trizol. Plates were gently scraped and Trizol/cell lysate was transferred to a 15 ml conical tube and stored at -80˚C until RNA isolation.

## RNA isolation: Adipocyte, stromal-vascular pellet, cultured cells

Samples in Trizol were removed from the -80˚C storage freezer and thawed at room temperature. Chloroform was added to each sample (10 ml to adipocyte, 5 ml to stromal-vascular pellet, 1 ml to cell culture), and tubes were recapped and vigorously shaken. Samples were then centrifuged at 12,000g for 15 minutes at 4˚C. Following centrifugation, the aqueous phase was removed and transferred to a fresh conical tube and an equivalent volume of 70% ethanol was added. Tubes were recapped and shaken. Samples were loaded onto RNeasy Maxi Kit (Qiagen Cat# 75152) columns and the RNeasy protocol for RNA washes and elution was followed (RNeasy Midi/Maxi Handbook, "Animal Tissues" Protocol beginning at step 7). RNA was eluted with 1.6 ml water. RNA was precipitated as follows: one-tenth volume 3M sodium acetate and 2.5X volume 100% ethanol were added to the 1.6 ml RNA sample. Tubes were capped, shaken to mix and stored overnight at -80˚C. Tubes were removed from -80˚C, thawed at room temperature and centrifuged at 16,000g for 20 minutes at 4˚C. Tubes were gently removed from the centrifuge, uncapped and quickly inverted to pour off liquid. These were left upside-down on paper towels to dry. Remaining ethanol droplets were aspirated and the RNA pellet was resuspended in 200 μl RNase-free water. RNA concentration was measured (Nanodrop ND-1000 spectrophotometer) and samples were stored at -80˚C.

## RNA isolation: Tissue

Tissue specimens were removed from the -80˚C freezer and thawed at room temperature. 20 ml Trizol was added to each sample. Tissue was homogenized in Trizol (Fisher Powergen 125 homogenizer) until no visible pieces of tissue remained. Five ml chloroform was added to each sample, tubes were recapped and vigorously shaken. Samples were then centrifuged at 12,000g for 15 minutes at 4˚C. Following centrifugation, RNA cleanup and precipitation was completed as described for adipose tissue fractions.

## RNA amplification

1 μg of each total RNA sample was amplified using Ambion Amino Allyl MessageAmp II-96 Kit (Cat# 1821) according to the manufacturer's protocol.

## DNA microarray production and prehybridization processing

HEEBO oligonucleotide microarrays, containing ~45,000 70-mer oligonucleotide probes, representing ~30,000 unique genes [102], were printed on epoxysilane-coated glass (Schott Nexterion E) by the Stanford Functional Genomics Facility.

Prior to hybridization, slides were first incubated in a humidity chamber (Sigma Cat# H6644) containing 0.5X SSC (1X SSC = 150 mM NaCl, 15 mM sodium citrate [pH 7.0]) for 30 minutes at room temperature. Slides were snap-dried at 70–80˚C on an inverted heat block. The free epoxysilane groups were blocked by incubation with 1M Tris-HCl (pH 9.0), 100 mM ethanolamine, and 0.1% SDS for 20 min at 50˚C. Slides were washed for 1 min in each of 2 wash chambers containing 400 ml of water, and then dried by centrifugation. Slides were used the same day.

## DNA microarray sample preparation, hybridization and washing

Amplified RNA was used for all DNA microarray experiments. Poly-adenylated RNAs were amplified in the presence of aminoallyl-UTP with Ambion Amino Allyl MessageAmp II-96 Kit (Cat# 1821). For mRNA expression experiments, Universal Reference RNA was used as an internal standard to enable reliable comparison of relative mRNA levels in multiple samples (Stratagene Cat# 740000) [103]. Ten μg each of amplified sample and reference RNA was fluorescently labeled with NHS-monoester Cy5 or Cy3 respectively (GE HealthSciences Cat# RPN5661). Differentially dye-labeled sample and reference RNA samples were combined, fragmented (Ambion Cat# 8740), then diluted into in a 50 μl solution containing 3X SSC, 25 mM Hepes-NaOH (pH 7.0), 20 μg of human Cot-1 DNA (Invitrogen Cat# 15279011), 20 μg of poly(A) RNA (Sigma Cat# P9403), 25 μg of yeast tRNA (Invitrogen Cat# 15401029), and 0.3% SDS. The sample was incubated at 90°C for 5 min, spun at 14,000 rpm for 10 min in a microcentrifuge, then hybridized at 65°C using the MAUI hybridization system (BioMicro) for 12–16 hours.

Following hybridization, microarrays were washed in a series of four solutions containing 400 ml of 2X SSC with 0.05% SDS, 2X SSC, 1X SSC, and 0.2X SSC, respectively. The first wash was performed for five minutes at 65°C. The subsequent washes were performed at room temperature for two minutes each. Following the last wash, the microarrays were dried by centrifugation in a low-ozone environment (<5 ppb) to prevent destruction of Cy dyes [104]. Once dry, the microarrays were kept in a low-ozone environment during storage and scanning.

## Microarray scanning and data processing

Microarrays were scanned using AxonScanner 4000B (Molecular Devices). PMT levels were adjusted manually to achieve minimal saturation. Each element was located and analyzed using SpotReader (Niles Scientific) or GenePix Pro 6.0 (Molecular Devices). The microarrays were submitted to the Stanford Microarray Database for further analysis [105]. Data were filtered to exclude elements that did not have: a regression correlation of $\geq 0.6$ between Cy5 and Cy3 signal over the pixels comprising the array element, and an intensity/background ratio of $\geq 2.5$ in at least one channel.

Microarray data from this study are available in GEO GSE25194.

## Generation of human tissue data (non-adipose tissues)

Tissues were collected by Stanford Tissue Bank and frozen at -80°C until RNA isolation. Tissue was thawed and 1 ml Trizol was added to each 50–100 mg specimen. Tissue was homogenized in Trizol (Fisher Powergen 125 homogenizer) until no visible pieces of tissue remained. 0.2 ml chloroform per ml of Trizol was added, tubes were recapped and vigorously shaken. Samples were then centrifuged at 12,000g for 15 minutes at 4°C. Aqueous phase was transferred to a new tube. To precipitate RNA, 0.5 ml isopropanol per ml Trizol was added, mixed well by pipetting and incubated for ten minutes at room temperature. Samples were spun at 12,000g for ten minutes at 4°C and supernatant was carefully discarded without disturbing the RNA pellet. To wash RNA, 75% ethanol was added to the RNA pellet and the sample was spun at 7500g for five minutes at 4°C. Supernatant was carefully discarded without disturbing the RNA pellet. RNA pellet was left at room temperature for five minutes to evaporate residual ethanol and dissolved in 50 μl RNase free water. RNA amplification, labeling and microarray hybridization experiments were carried out as described above for adipose tissue specimens.

## RT-qPCR

Five mg of total RNA from each sample was reverse transcribed with random hexamers at 25°C for 10 min then 37°C for 2 hrs (High Capacity cDNA Archive Kit, cat # 4368813) in 150 μl reactions. Taqman probe sets for fifty mRNAs (S11 Dataset) were diluted to 5X working concentrations and 5 μl were aliquoted into 96-well plates (ABI cat # 4346906) by robot (Velocity11 VPrep) at the Stanford High-Throughput Bioscience Center and stored at -20°C. For each qPCR reaction, 60 μl of the reverse transcription reaction was mixed with 600 μl of TaqMan Gene Expression Master Mix (ABI cat # 4370074) and 540 μl of water; 20 μl of the resulting mixture was added to each of the 50 primer combinations in the 96-well plates and mixed. We transferred 2 x 10 μl from each well into a 384-well plate (ABI cat # 4309849), covered the plates (ABI cat # 4306311), then centrifuged the plates at 3,000 rpm for 10 minutes in a Beckman tabletop centrifuge and stored the samples at -20°C for up to two weeks. qPCR reactions were carried out in an ABI 7900T: 95°C/10min, followed by 50 cycles at 95°C/15 sec and 60°C/60 sec. Default settings were used for Ct calculations. Flagged measurements were removed prior to taking the average of technical replicates and the averaged Ct value for each transcript was normalized to PPIA expression (assay id # Hs99999904_m1), which was chosen as the normalization control because its expression does not vary considerably across adipocyte samples, as measured by HEEBO arrays.

## Microarray analyses

The *in vivo* and *ex vivo* datasets were processed separately. Following extraction of the dataset from the Stanford Microarray Database, we performed quality assessment of the dataset and removed outlier arrays and corrected a handful of mislabeled samples. The quality filtered datasets and associated manifests are provided in S1–S4 Datasets.

The *in vivo* dataset was further processed by removing microarray probes with missing data in >25% of samples and imputing missing data with impute R package (knn = 10). This dataset was used for analyses in Fig 1. The adipocyte dataset was obtained by removing probes in the bottom quartile of expression in adipocytes vs. stromal cells (SAM unpaired t-test), then subtracting the average value for each probe to obtain mean-centered data. This dataset was used for subsequent analyses, with two exceptions: we manually added back PPARG2 probe (hHA034694) for analyses related to Fig 3 (filtered out because it had missing data in >25% of all *in vivo* samples), and we manually added back the FOXD1 probe (hHC007832) for analyses related to Fig 5 (filtered out because it was in 19th percentile in adipocyte vs. stromal cell expression).

The *ex vivo* dataset was further processed by removing microarray probes with missing data in >40% of samples and imputing missing data with impute R package (knn = 10). This dataset was used for analyses in Fig 6.

Clustering was performed with Cluster 3.0 for Mac OS X [106] and visualized with Java TreeView 1.1.1-osx [107]. Principal component analyses were performed using the R package "irlba". Unpaired two-class *t*-tests and multi-class comparisons were performed using SAM [42] run in R (samr) with default settings with 1000 permutations of the data to generate estimated FDRs. Enrichment of GO terms was performed with a custom R script utilizing the package "GSA" and p-values were corrected for multiple-hypothesis testing using Benjamini-Hochberg Procedure. GSEA was performed using "fgsea" R package. PAM [108] was run in R (pamr), with default settings. To estimate the significance of HOX gene expression predicting site of origin, we first performed PAM with HOX genes. We then ran PAM on 10,000 randomly chosen sets of 28 mRNAs and found that the HOX PAM score was higher than the scores for all 10,000 randomly chosen sets.

The R package "tidyverse" was utilized for data manipulation and plotting [109].

### GTex RNA sequencing analyses

Adipose tissue GTex V8 RNA sequencing data were retrieved and processed using the "recount3" R package [110]. We transformed the data into normalized counts per gene using the "TMM" method, filtered genes by expression level using "filterByExpr", and performed statistical comparisons between depot and sexes using limma package "voom", "lmFit" and "eBays" commands [111].

### Gene sets

Gene Ontology biological process (c5.go.bp.v7.4.symbols.gmt) and molecular function (c5.go.mf.v7.5.1.symbols.gmt) annotations were obtained from The Molecular Signatures Database [48]. Putative PPARγ targets were obtained from [62, 63]. Developmental transcription factors (Fig 4) were defined as the intersection of "GOMF_DNA_BINDING_TRANSCRIPTION_FACTOR_ACTIVITY" with "GOBP_ANIMAL_ORGAN_MORPHOGENESIS|GOBP_EMBRYO_DEVELOPMENT|GOBP_CELL_FATE_COMMITMENT|GOBP_ANATOMICAL_STRUCTURE_FORMATION_INVOLVED_IN_MORPHOGENESIS".

## Supporting information

**S1 Table. Detailed information for adipocyte samples collected.** List of adipocyte sample information including adipose depot of origin, patient gender and age, date of collection/adipocyte isolation, and any other known information such as surgical procedure. F = female, M = male, followed by the age of the patient; U denotes either unknown patient sex or age. Lowercase letters are used to distinguish patients with the same sex and age. In the notes column "not OW/obese" indicates that the patient was not overweight or obese.
(DOCX)

**S2 Table. Concordance between microarray and RT-qPCR results.** For each gene shown in S7–S10 Figs, the table lists the Pearson correlation and associated p-value between that gene's relative expression measured by microarray vs. RT-qPCR, using the 46 samples profiled on both platforms.
(DOCX)

**S1 Fig. Anatomic location of adipose depots chosen for this study.** Subcutaneous depots were defined as fat found just underneath the skin: abdominal subcutaneous ranging from anywhere underneath the skin on the ventral side of the body between the rib cage and pelvis, thigh ranging anywhere from the hip to 2 inches above the knee spanning the entire circumference of the leg, shin/calf anywhere below the knee but above the ankle spanning the circumference of the leg. Breast fat samples were either directly subcutaneous, interdigitated with the mammary glands, or a combination of these. Omental and pericolonic depots are both located within the peritoneal cavity (see cross section). Omental fat is a component of the omentum, consisting of a mesothelial pouch that encloses adipose tissue embedded in loose connective tissue and interspersed with aggregates of phagocytic cells termed "milky spots". The omentum hangs like an apron from the stomach, covering the abdominal organs connecting with the transverse colon and posterior abdominal wall. Pericolonic or epiploic appendices are small pouches of peritoneum filled with fat and situated along the outer surface of the colon and upper part of the rectum. Retroperitoneal fat was excised from the retroperitoneal space, the compartment behind the peritoneum that houses the kidneys, bladder and adrenal glands (see cross section); the surgical samples were labeled "perinephric", "perirenal", or simply

"retroperitoneal".
(TIF)

**S2 Fig. Expression patterns of mRNAs that vary among adipose depots segregate sites by anatomic location and have distinct functional themes.** Supervised hierarchical clustering of mRNAs whose variation in expression is largely explained by inter-depot differences (multi-class SAM FDR < 0.2%) and whose expression in adipocytes or ASCs was affirmed by single cell RNA sequencing [58].
(TIF)

**S3 Fig. Variation in expression levels of gene sets marking distinct adipocyte populations identified by single cell sequencing.** Boxplot representations of the relative expression across depots for seven gene sets that mark distinct adipocyte populations according to [58]. hAD1,3,4,5 and 7 were identified from abdominal subcutaneous adipocytes and hAD2 and 6 were identified from omentum and peritoneal visceral adipocytes.
(TIF)

**S4 Fig. Transcription factors linked to lipid metabolism with depot-specific expression patterns.** (A-F) Boxplot representation of the relative expression of the indicated gene across depots.
(TIF)

**S5 Fig. HOX gene expression marks adipocytes by depot of origin and may provide intrinsic positional identity.** Supervised hierarchical clustering of HOX genes across adipocytes.
(TIF)

**S6 Fig. Robust induction of PPARγ targets following *ex vivo* differentiation of ASCs.** Boxplots of the relative fold-change of PPARγ targets in EVD cells vs. ASCs. Each point represents the average fold-change of the PPARγ targets from one sample. To calculate the fold-change for each mRNA following differentiation, the average $\log_2$ expression value across ASC samples from that site was subtracted from the $\log_2$ EVD adipocyte expression value.
(TIF)

**S7 Fig. Developmental transcription factor expression in adipocytes as measured by microarray and RT-qPCR.** (A-L) Boxplots of the relative expression of the indicated gene in each site according to microarray (left) and RT-qPCR (right).
(TIF)

**S8 Fig. Expression of putative PPARγ1 targets in adipocytes as measured by microarray and RT-qPCR.** (A-F) Boxplots of the relative expression of the indicated gene in each site according to microarray (left) and RT-qPCR (right).
(TIF)

**S9 Fig. Expression of putative PPARγ2 Targets in adipocytes as measured by microarray and RT-qPCR.** (A-L) Boxplots of the relative expression of the indicated gene in each site according to microarray (left) and RT-qPCR (right).
(TIF)

**S10 Fig. Expression of adiponectin and leptin in adipocytes as measured by microarray and RT-qPCR.** (A) Boxplots of the relative expression of ADIPOQ in each site according to microarray (left) and RT-qPCR (right). (B) Boxplots of the relative expression of LEP in each site according to microarray (left) and RT-qPCR (right). (C) Scatterplot comparing the relative expression of adiponectin (ADIPOQ) and leptin (LEP) across samples, as measured by RT-

qPCR. Sample depot is color-coded.
(TIF)

**S1 Dataset. Manifest for *in vivo* microarray dataset.**
(TXT)

**S2 Dataset. Unprocessed *in vivo* microarray dataset.**
(TXT)

**S3 Dataset. Manifest for *ex vivo* microarray dataset.**
(TXT)

**S4 Dataset. Unprocessed *ex vivo* microarray dataset.**
(TXT)

**S5 Dataset. *in vivo* overview cluster Fig 1.**
(TXT)

**S6 Dataset. Adipocyte overview cluster Fig 2.**
(TXT)

**S7 Dataset. GO results overview cluster Fig 2.**
(TXT)

**S8 Dataset. PPARγ targets expression profiles Fig 3.**
(TXT)

**S9 Dataset. HOX expression profiles Fig 4 and S5 Fig.**
(TXT)

**S10 Dataset. Developmental transcription factor expression profiles Fig 5.**
(TXT)

**S11 Dataset. RT-qPCR values relative to normalization control PP1A.**
(TXT)

## Acknowledgments

We thank Dr. Erica Dobo at Stanford Surgical Pathology for working with us to procure the majority of the adipose tissue samples used for this research. We also thank Stanford Hospital and Clinics surgeons, Drs. Hayes Gladstone, David Kahn, Irene Wapnir, and Mark Welton, as well as the Stanford Tissue Bank for aiding in sample collection. We thank Robert Pesich for supplying the non-adipose tissue microarray data.

## Author Contributions

**Conceptualization:** Patrick O. Brown.

**Data curation:** Heather J. Clemons, Daniel J. Hogan.

**Formal analysis:** Heather J. Clemons, Daniel J. Hogan.

**Funding acquisition:** Patrick O. Brown.

**Investigation:** Heather J. Clemons, Daniel J. Hogan, Patrick O. Brown.

**Methodology:** Heather J. Clemons, Patrick O. Brown.

**Project administration:** Heather J. Clemons, Patrick O. Brown.

**Resources:** Patrick O. Brown.

**Supervision:** Patrick O. Brown.

**Visualization:** Heather J. Clemons.

**Writing – original draft:** Heather J. Clemons, Daniel J. Hogan.

**Writing – review & editing:** Daniel J. Hogan, Patrick O. Brown.

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
