## [Decision Letter · Decision Letter 0]

11 Apr 2024

PONE-D-24-04146Depot-Specific mRNA Expression Programs in Human Adipocytes Suggest Physiological Specialization via Distinct Developmental ProgramsPLOS ONE

Dear Dr. Hogan,

Thank you for submitting your manuscript to PLOS ONE. After careful consideration, we feel that it has merit but does not fully meet PLOS ONE’s publication criteria as it currently stands. Therefore, we invite you to submit a revised version of the manuscript that addresses the points raised during the review process.

We look forward to receiving your revised manuscript.

Kind regards,

Licy Yanes Cardozo

Academic Editor

PLOS ONE

Journal Requirements:

"This work was supported by the Howard Hughes Medical Institute and by a grant from the NIH to POB, “Extending and Interpreting Molecular Portraits of Cancer” (NIH 5 R01 CA111487). HJC was partially supported by the Stanford Genome Training Program (Grant Number 5 T32 HG000044)."

"This work was supported by the Howard Hughes Medical Institute and by a grant from the NIH

to POB, “Extending and Interpreting Molecular Portraits of Cancer” (NIH 5 R01 CA111487). HJC

was partially supported by the Stanford Genome Training Program (Grant Number 5 T32

HG000044)."

"This work was supported by the Howard Hughes Medical Institute and by a grant from the NIH to POB, “Extending and Interpreting Molecular Portraits of Cancer” (NIH 5 R01 CA111487). HJC was partially supported by the Stanford Genome Training Program (Grant Number 5 T32 HG000044)."

6. We note that your Data Availability Statement is currently as follows: Microarray data from this study are available in GEO GSE25194. Datasets used to generate main figures will be available as supplemental material.

7. We notice that your supplementary figures/tables are included in the manuscript file. Please remove them and upload them with the file type 'Supporting Information'. Please ensure that each Supporting Information file has a legend listed in the manuscript after the references list.

**Additional Editor Comments:**

Please addresses comments that rose from both reviewers. Thanks 

Reviewers' comments:

Reviewer's Responses to Questions

**Comments to the Author**

1. Is the manuscript technically sound, and do the data support the conclusions?

Reviewer #1: Yes

Reviewer #2: Yes

2. Has the statistical analysis been performed appropriately and rigorously? 

Reviewer #1: Yes

Reviewer #2: Yes

3. Have the authors made all data underlying the findings in their manuscript fully available?

Reviewer #1: Yes

Reviewer #2: Yes

4. Is the manuscript presented in an intelligible fashion and written in standard English?

Reviewer #1: Yes

Reviewer #2: Yes

5. Review Comments to the Author

Reviewer #1: Clemons et al. reported the global transcriptomic analysis of human adipose tissue obtained from several different depots depots by using DNA microarray. In addition, authors also compared the gene profile between adipose tissue, adipose tissue derived stem cells (preadipocytes), and ex vivo differentiated adipocytes.

I would like to address several points to authors:

1. In Figure 1B, authors showed heatmap for adipocyte, stromal vascular, and adipose tissue from several depots. Please clarify from which depots. Is this data correlated with Figure 2?

2. Authors included GPAM as canonical adipocyte marker, please include the reference.

3. The description of Figure 2B-F data needs a clarification. Box plots showed the relative expression of mRNAs in each cluster, does this data comprise all of mRNAs in these clusters? If these are relative expression, what was the reference used for the comparison? Please clarify this point.

4. Including supplementary tables listing all genes involved in each overrepresented pathways showed in Figure 2B-F is advised.

5. The differentiation protocol for ex vivo differentiated adipocytes was not found in the methods. Please clarify the differentiation protocol and reference.

6. Heatmap scale legend is missing in all of heatmaps. Only by looking at the heatmap, readers can not find the information about the color scale. Although the meaning of the color is written in the figure legends.

7. For Figure 6, when authors compared the gene profile of in vivo adipocytes, ASCs, and EVD from 5 different tissues, were the samples obtained from similar individuals to avoid interindividual difference? Did authors take in vivo adipocytes, ASCs, and EVD from one individual?

8. Since authors showed in Figure 1A that both SVF and ASCs were collected, I am curious whether authors compared the gene profile between SVF and ASCs? Is there any differentially expressed genes between two groups?

9. In the discussion, authors nicely explained the medical perspective of their findings, citing a genetic variant in an enhancer for IRX3. Previous findings have reported that risk allele of FTO 1421085 SNP increased the expression of IRX3 and 5, thus increased the fat storage. In addition, some studies in human abdominal subcutaneous adipocytes has also been reported related to FTO 1421085 SNP. Authors showed that IRX1, 2, 3, and 5 were abundant in abdominal, thigh, and calf, but low in omentum and pericolonic. Does this mean that the expression of IRXs are depot-specific regardless of the FTO SNP?

10. This study comprehensively reported the transcriptomic profile of several adipose depots, however there was no validation by qPCR or western blot found for selected or important markers, which can be considered as the limitation of this study.

Reviewer #2: General Critique: This is a well-written manuscript that assesses differential gene expression programs between not only visceral versus subcutaneous fat but various anatomic locations, including omentum, pericolonic, retroperitoneal, and abdominal, breast, shin, and thigh subcutaneous areas. The premise of the work is that the location of adipose tissue will be inherently programmed during fetal life and will make the tissue gene expression suite similar to the surrounding tissues (ie, skeletal muscle, liver etc) but dissimilar to the gene expression suite of adipose tissue located in other areas. This premise is supported by the similarity between gene expression in the stromal vascular fraction (SVF) and induced adipocytes in cell culture. This paper is exceptionally comprehensive, and the data analysis overall is appropriate and thorough. However, one very significant issue with the paper is the structure in that much discussion and references about individual genes or findings occur in the results rather than the discussion section. This makes the discussion somewhat vague and less detailed and also breaks up the results section. This reviewer suggests that all discussion and references be moved into the discussion section and be reworked for the flow of the manuscript. Additionally, the authors switch between present and past tense in the discussion of the literature in the introduction. This reviewer suggests using present tense (ie, “X study finds abcd”). The abstract is also very vague as to the main findings of the manuscript. The abstract should be reworked to mention the main driving points of the manuscript more specifically. One drawback to the data analysis and the discussion also is the pooling of data from both sexes. It is well-known that sex (i.e., XX vs. XY) and sex steroids play a role in the propensity for adipose tissue deposition in specific subcutaneous adipose tissue locations as well as viscerally. This is completely ignored in the data set. The authors do have the sex and age of the patient samples. It is the opinion of this reviewer that the authors need to address sex*location interaction in the data set up front first thing. If they can demonstrate no significant interaction between these two variables and share that data as a supplemental figure or table then they can proceed with the pooled-sex analysis as they have already done. Specific critiques are as follows:

Specific Critiques:

1) Results: Although the manuscript line numbers are not indicated, under the Results section entitled “Maintenance of site-specific differences ex vivo” there is a section of text that has already been mentioned in the M&M. The authors should remove this text from the Results section.

2) Discussion: don’t refer back to figures specifically in the discussion

3) Figures:

a. Figure 2: the section under (A) is already mentioned verbatim in the text. Pick a spot where you want this text and put it either in the results text OR in the figure legend. Not both. The figure legend text for B-F is redundant; reword so that you are not saying “same as (B) except for…” for each panel.

b. Figure 5: Streamline the figure legend text for B-J so it is not redundant (see comment for Figure 2 above). Although it may be somewhat self-evident, define the abbreviations in panels B-J (ie, A, B, O etc) in the figure legend.

c. Figure 6 B and C: this reviewer is really underwhelmed by the presentation of data in panels B and C here. I feel like there is a more impactful way to show the correlation of the top 200 mRNAs. Please explain why you chose this approach or rework the data to show panels in this figure that are more impactful. Ultimately, the omentum has the worst adipocyte induction in culture, so there is the least gene variation between the 3 sampling conditions. The shin appears to vary more so.

6. PLOS authors have the option to publish the peer review history of their article (what does this mean?). If published, this will include your full peer review and any attached files.

Reviewer #1: No

Reviewer #2: No

---

## [Author Response · Author response to Decision Letter 0]

26 May 2024

We updated the manuscript in accordance with PLoS One guidelines and also updated the figures using PACE.

We didnt use LaTeX.

"This work was supported by the Howard Hughes Medical Institute and by a grant from the NIH to POB, “Extending and Interpreting Molecular Portraits of Cancer” (NIH 5 R01 CA111487). HJC was partially supported by the Stanford Genome Training Program (Grant Number 5 T32 HG000044)."

"This work was supported by the Howard Hughes Medical Institute and by a grant from the NIH

to POB, “Extending and Interpreting Molecular Portraits of Cancer” (NIH 5 R01 CA111487). HJC

was partially supported by the Stanford Genome Training Program (Grant Number 5 T32

HG000044)."

"This work was supported by the Howard Hughes Medical Institute and by a grant from the NIH to POB, “Extending and Interpreting Molecular Portraits of Cancer” (NIH 5 R01 CA111487). HJC was partially supported by the Stanford Genome Training Program (Grant Number 5 T32 HG000044)."

"This work was supported by the Howard Hughes Medical Institute and by a grant from the NIH to POB, “Extending and Interpreting Molecular Portraits of Cancer” (NIH 5 R01 CA111487). HJC was partially supported by the Stanford Genome Training Program (Grant Number 5 T32 HG000044)."

6. We note that your Data Availability Statement is currently as follows: Microarray data from this study are available in GEO GSE25194. Datasets used to generate main figures will be available as supplemental material.

All data necessary to reproduce the results are provided in GEO and supplemental datasets.

7. We notice that your supplementary figures/tables are included in the manuscript file. Please remove them and upload them with the file type 'Supporting Information'. Please ensure that each Supporting Information file has a legend listed in the manuscript after the references list.

Done

Reviewer #1: Clemons et al. reported the global transcriptomic analysis of human adipose tissue obtained from several different depots depots by using DNA microarray. In addition, authors also compared the gene profile between adipose tissue, adipose tissue derived stem cells (preadipocytes), and ex vivo differentiated adipocytes.

I would like to address several points to authors:

1. In Figure 1B, authors showed heatmap for adipocyte, stromal vascular, and adipose tissue from several depots. Please clarify from which depots. Is this data correlated with Figure 2?

In general, from each biopsy/patient, we profiled the whole tissue, the floated adipocytes, the pelleted stromal-vascular cells, cultured adipose stem cells from the stromal-vascular cells, and ex vivo differentiated adipocytes from the cultured adipose stem cells. So Figure 1 includes the adipocyte data presented subsequently in the manuscript, including Figure 2. The point of figure 1 is that the tissue fraction was the major source of variation, which is why we did not color code the depot of origin (or other info, such as procedure, or sex). 

2. Authors included GPAM as canonical adipocyte marker, please include the reference.

GPAM, also known as GPAT1, catalyzes the first committed step in triacylglycerol synthesis. We included a citation showing it is expressed specifically in adipocytes (PMID: 34380013).

3. The description of Figure 2B-F data needs a clarification. Box plots showed the relative expression of mRNAs in each cluster, does this data comprise all of mRNAs in these clusters? If these are relative expression, what was the reference used for the comparison? Please clarify this point.

The microarray data were normalized by (1) log2 transformation, (2) mean centering each sample, and then (3) mean centering each mRNA. For the boxplots, we used all the data points used to make the heatmap. So for cluster 1 (2B), the boxplot for abdominal uses all the values from each mRNA in cluster 1 (71 mRNAs) for all abdominal samples (11), and so on. The bottom and top of the box represent the 1st and 3rd quartiles. The top line extends from the top of the box to the largest value no further than 1.5 times the interquartile range (IQR) from the box; the bottom line extends from the rectangle to the smallest value at most 1.5 times IQR of the box. The horizontal bar in the box is the median.

4. Including supplementary tables listing all genes involved in each overrepresented pathways showed in Figure 2B-F is advised.

We added a supplemental table with this information – S7 Dataset.

5. The differentiation protocol for ex vivo differentiated adipocytes was not found in the methods. Please clarify the differentiation protocol and reference.

The protocol was/is provided in Materials and Methods sections “Cell Culture” and “Adipocyte Differentiation”.

Cell Culture

One-half of each stromal-vascular pellet fraction from the “Tissue Fractionation” procedure was reserved to culture adipose stem cells. The cell suspension was centrifuged at 400g for 7 minutes and the buffer was carefully aspirated with a pipette to avoid losing pelleted cells. Cells originating from the pericolonic or omental depots were additionally treated for 10 min with RBC lysis buffer*, again centrifuged, and the lysis buffer aspirated. Cells were resuspended in 10 ml DMEM + 10% FBS + 2.5 μg/ml Fungizone (Invitrogen Cat# 15290-018) + Pen/Strep/Glutamine (Invitrogen 10378-016). Cells were transferred to the appropriate size culture flask (10-20 g starting tissue in 1 x T25, 20-50g in 1x T75). Additional DMEM (~1ml/5cm2) was added to cells and mixed by pipetting. Cells were incubated overnight or up to 24 hours to allow time for adipose stem cells to adhere. Cells were rinsed with warm PBS and given fresh medium. Presence of adherent stromal cells was assessed visually at 10x magnification. Successful cultures were continually grown in DMEM + 10%FBS + Pen/Strep/Glutamine. Fungizone was no longer added after 2 passages. Cells were split 1 to 3 at 70-90% confluence. When enough cells were obtained (at least 1 confluent T225 flask), cells were rinsed with PBS and trypsinized, pelleted by centrifugation and resuspended in fresh medium. Each of 2, 10 cm dishes were plated, one for pre- and one for post-differentiation RNA collection. For the latter see “Adipocyte Differentiation”. Cells for pre-differentiation RNA collection were plated at 100% confluence in fresh DMEM and incubated overnight. Cells were photographed at 10x magnification to document confluence and morphology of cells. DMEM was aspirated and cells were rinsed with warm PBS. PBS was replaced by 6 ml Trizol. Plates were gently scraped and Trizol/cell lysate was transferred to a 15 ml conical tube and stored at -80°C. 

*RBC lysis buffer: In 1 liter sterile water, 8.26 g ammonium chloride (NH4Cl), 1 g potassium bicarbonate (KHCO3), and 0.037 g EDTA. 

Adipocyte Differentiation

Adipose stem cells were grown to confluence as described in “Cell Culture”. Cells were rinsed with warm PBS and trypsinized. Trypsinized cells were transferred to a 15 ml conical tube, pelleted by centrifugation and resuspended in 5 ml DMEM + 10% FBS + Pen/Strep/Glutamine (Invitrogen Cat# 10378-016). Cells were plated to achieve the greatest density possible depending on the number of cells available, as higher confluence yielded a greater percentage of differentiating cells. In general, cells were plated at 100-300% confluence. Cells were incubated overnight and photographed the following day at 10x magnification to document confluence and record the morphology of cells. Cells were rinsed with warm PBS. PBS was replaced with Zen-Bio differentiation medium (Cat# DM-2) and the differentiation procedure was followed as described in Zen-Bio instruction manual ZBM-1. At 16 days following induction of differentiation, cells were photographed at 10x magnification to document the number and morphology of differentiating cells. Cells were rinsed with warm PBS. PBS was replaced with 6 ml Trizol. Plates were gently scraped and Trizol/cell lysate was transferred to a 15 ml conical tube and stored at -80°C until RNA isolation. 

6. Heatmap scale legend is missing in all of heatmaps. Only by looking at the heatmap, readers can not find the information about the color scale. Although the meaning of the color is written in the figure legends.

We have now provided heatmap scales in the figures.

7. For Figure 6, when authors compared the gene profile of in vivo adipocytes, ASCs, and EVD from 5 different tissues, were the samples obtained from similar individuals to avoid interindividual difference? Did authors take in vivo adipocytes, ASCs, and EVD from one individual?

In general, from each biopsy/patient, we profiled the whole tissue, the floated adipocytes, the pelleted stromal-vascular cells, cultured adipose stem cells from the stromal-vascular cells, and ex vivo differentiated adipocytes from the cultured adipose stem cells. 

8. Since authors showed in Figure 1A that both SVF and ASCs were collected, I am curious whether authors compared the gene profile between SVF and ASCs? Is there any differentially expressed genes between two groups?

We did not perform these analyses. But we’ve made the data required for this analysis (and many other comparisons that readers might find interesting) available in GEO and supplemental datasets.

9. In the discussion, authors nicely explained the medical perspective of their findings, citing a genetic variant in an enhancer for IRX3. Previous findings have reported that risk allele of FTO 1421085 SNP increased the expression of IRX3 and 5, thus increased the fat storage. In addition, some studies in human abdominal subcutaneous adipocytes has also been reported related to FTO 1421085 SNP. Authors showed that IRX1, 2, 3, and 5 were abundant in abdominal, thigh, and calf, but low in omentum and pericolonic. Does this mean that the expression of IRXs are depot-specific regardless of the FTO SNP?

That is correct. Variation in expression of IRX1,2,3 and 5 is mostly depot-dependent (and likely programmed during mesoderm development). That being said, the FTO SNP could certainly augment expression in specific depots, as it is believed to be in an IRX3 enhancer.

10. This study comprehensively reported the transcriptomic profile of several adipose depots, however there was no validation by qPCR or western blot found for selected or important markers, which can be considered as the limitation of this study.

We do in fact have RT-qPCR data for 50 mRNAs for 51 of 59 samples for which we have microarray data, plus 26 additional samples. We added this information to the paper (lines, S7-10 Figures) and have also provided the qPCR dataset (S11 dataset). 

Reviewer #2: General Critique: This is a well-written manuscript that assesses differential gene expression programs between not only visceral versus subcutaneous fat but various anatomic locations, including omentum, pericolonic, retroperitoneal, and abdominal, breast, shin, and thigh subcutaneous areas. The premise of the work is that the location of adipose tissue will be inherently programmed during fetal life and will make the tissue gene expression suite similar to the surrounding tissues (ie, skeletal muscle, liver etc) but dissimilar to the gene expression suite of adipose tissue located in other areas. This premise is supported by the similarity between gene expression in the stromal vascular fraction (SVF) and induced adipocytes in cell culture. This paper is exceptionally comprehensive, and the data analysis overall is appropriate and thorough. However, one very significant issue with the paper is the structure in that much discussion and references about individual genes or findings occur in the results rather than the discussion section. This makes the discussion somewhat vague and less detailed and also breaks up the results section. This reviewer suggests that all discussion and references be moved into the discussion section and be reworked for the flow of the manuscript. Additionally, the authors switch between present and past tense in the discussion of the literature in the introduction. This reviewer suggests using present tense (ie, “X study finds abcd”). The abstract is also very vague as to the main findings of the manuscript. The abstract should be reworked to mention the main driving points of the manuscript more specifically. 

We corrected inconsistencies in tense usage. We did not make substantial changes to the abstr

---

## [Decision Letter · Decision Letter 1]

20 Aug 2024

PONE-D-24-04146R1Depot-Specific mRNA Expression Programs in Human Adipocytes Suggest Physiological Specialization via Distinct Developmental ProgramsPLOS ONE

Dear Dr. Hogan,

Thank you for submitting your manuscript to PLOS ONE. After careful consideration, we feel that it has merit but does not fully meet PLOS ONE’s publication criteria as it currently stands. Therefore, we invite you to submit a revised version of the manuscript that addresses the points raised during the review process.

Well written review, please add review and complete the statistical analysis of the new results as requested by reviewer. Please updated figures accordingly ==============================

We look forward to receiving your revised manuscript.

Kind regards,

Licy Yanes Cardozo

Academic Editor

PLOS ONE

Journal Requirements:

**Additional Editor Comments:**

well written article, please addressed concerns risen by Reviewer 3

Reviewers' comments:

Reviewer's Responses to Questions

**Comments to the Author**

1. If the authors have adequately addressed your comments raised in a previous round of review and you feel that this manuscript is now acceptable for publication, you may indicate that here to bypass the “Comments to the Author” section, enter your conflict of interest statement in the “Confidential to Editor” section, and submit your "Accept" recommendation.

Reviewer #1: All comments have been addressed

Reviewer #3: (No Response)

2. Is the manuscript technically sound, and do the data support the conclusions?

Reviewer #1: Yes

Reviewer #3: Partly

3. Has the statistical analysis been performed appropriately and rigorously? 

Reviewer #1: Yes

Reviewer #3: No

4. Have the authors made all data underlying the findings in their manuscript fully available?

Reviewer #1: Yes

Reviewer #3: Yes

5. Is the manuscript presented in an intelligible fashion and written in standard English?

Reviewer #1: Yes

Reviewer #3: Yes

6. Review Comments to the Author

Reviewer #1: Authors have addressed all of the points from previous review round.

I may miss it, but authors may clarify how the mechanistic figures in Fig. 1A was prepared.

If authors used a platform that requires license, please provide it.

Reviewer #3: This study reports the transcriptomic profile of various anatomically distinct adipose depots, highlighting potential depot-specific characteristics likely resulting from intrinsic developmental programs. The study is comprehensive and provides extensive data that could enhance our understanding of the diverse functionality of these depots. Below are some suggestions for the authors:

In Figures 1C, 5B-J, and 7B-C, it is crucial to show the statistical analysis results (significance) for the relative gene expression between different depots. This should be highlighted both in the figures and the text description.

Please also indicate the significance for the RT-qPCR data presented in Figures S7, S8, S9, and S10 (A, B).

7. PLOS authors have the option to publish the peer review history of their article (what does this mean?). If published, this will include your full peer review and any attached files.

Reviewer #1: No

Reviewer #3: No

---

## [Decision Letter · Decision Letter 2]

24 Sep 2024

Depot-Specific mRNA Expression Programs in Human Adipocytes Suggest Physiological Specialization via Distinct Developmental Programs

PONE-D-24-04146R2

Dear Dr. Hogan,

We’re pleased to inform you that your manuscript has been judged scientifically suitable for publication and will be formally accepted for publication once it meets all outstanding technical requirements.

Kind regards,

Licy Yanes Cardozo

Academic Editor

PLOS ONE

Additional Editor Comments (optional):

All the concerns made by reviewers were addressed and this manuscript is acceptable for publication

Reviewers' comments:

Reviewer's Responses to Questions

**Comments to the Author**

1. If the authors have adequately addressed your comments raised in a previous round of review and you feel that this manuscript is now acceptable for publication, you may indicate that here to bypass the “Comments to the Author” section, enter your conflict of interest statement in the “Confidential to Editor” section, and submit your "Accept" recommendation.

Reviewer #3: All comments have been addressed

2. Is the manuscript technically sound, and do the data support the conclusions?

Reviewer #3: Yes

3. Has the statistical analysis been performed appropriately and rigorously? 

Reviewer #3: Yes

4. Have the authors made all data underlying the findings in their manuscript fully available?

Reviewer #3: Yes

5. Is the manuscript presented in an intelligible fashion and written in standard English?

Reviewer #3: Yes

6. Review Comments to the Author

Reviewer #3: The authors have adequately addressed the comments, and the manuscript is acceptable in its current form.

7. PLOS authors have the option to publish the peer review history of their article (what does this mean?). If published, this will include your full peer review and any attached files.

Reviewer #3: No

---

## [Editor Report · Acceptance letter]

3 Oct 2024

PONE-D-24-04146R2 

PLOS ONE

Dear Dr. Hogan, 

I'm pleased to inform you that your manuscript has been deemed suitable for publication in PLOS ONE. Congratulations! Your manuscript is now being handed over to our production team.

Kind regards, 

on behalf of

Dr. Licy Yanes Cardozo 

Academic Editor

PLOS ONE